# Epitope Generation for Peptide-based Cancer Vaccine using Goal-directed Wasserstein Generative Adversarial Network with Gradient Penalty

## Abstract

We introduce a novel goal-directed Wasserstein Generative Adversarial Network with Gradient Penalty (GD-WGAN-GP) for training a generator capable of producing peptide sequences with high predicted immunogenicity and strong binding affinity to the human leukocyte antigen HLA-A*0201, thereby eliciting cytotoxic T-cell immune responses. The proposed GD-WGAN-GP incorporates a critic network to guide the generator in producing peptides with a strong binding affinity similar to those in the training set and a reward network to steer the generator toward producing sequences with high predicted immunogenicity. To avoid the generator prioritizing the objective of the critic at the expense of immunogenicity, we introduce a scaling factor to balance the influence of the reward in the loss of the generator. To reduce peptide repetition, we integrate the reward into the loss of the generator using two approaches: a switching mechanism that excludes the reward term when duplicated peptides are present in a batch, and otherwise multiplies it by a $\gamma_{max}$ parameter to control the reward's contribution, and (2) a repetition penalty from ORGAN, where each reward is divided by the number of occurrences of its corresponding peptide within the batch. Experiments on bladder cancer epitope sequences demonstrate that GD-WGAN-GP with the switching mechanism enables a tunable trade-off between the number of unique peptides and the average immunogenicity score via varying $\gamma_{max}$. Furthermore, the generator trained by the GD-WGAN-GP with the ORGAN's repetition penalty achieves an optimal balance of uniqueness and immunogenicity. Across multiple datasets, GD-WGAN-GP outperforms existing methods by effectively reducing peptide redundancy while preserving high immunogenicity scores and strong binding affinity. The Python codes are provided at: `https://github.com/AnnonymousForPapers/GP-WGAN-GP_with_switch_and_ORGAN_penalty`.

## 1 Introduction

Tumor-specific antigens have been utilized in cancer vaccines, a form of cancer immunotherapy, to stimulate the production of tumor-specific T cells (Stevanovic, 2002). These antigens are encoded in the genome and are not present in normal cells, making them representative of aberrant proteins (Gubin et al., 2015). An epitope can be defined as a segment of an antigen that is generated through antigen processing (Sidney et al., 2020). The activation of the tumor-specific T-cell response occurs when epitopes from class I or class II human leukocyte antigen (HLA) molecules are presented to T cell receptors (TCRs), which recognize antigens in the form of peptides (Mohme & Neidert, 2020). The HLA class I molecule serves as the TCR ligand for CD8+ cytotoxic T lymphocytes (CTLs) and binds peptides consisting of 8-11 amino acids. On the other hand, the TCR ligand for CD4+ helper T cells is the HLA class II molecule, and the bound peptides typically consist of 15 amino acids (Sim & Sun, 2022).

Peptide-based cancer vaccines refer to cancer vaccines that employ polypeptides comprised of known or predicted tumor antigen epitopes (Liu et al., 2022). Peptide-based vaccines are widely utilized in cancer vaccination practices and are designed to activate a tumor-specific cytotoxic T lymphocyte (CTL) response (Butterfield, 2015). These vaccines consist of multiple epitopes, typically ranging from 8 to 11 amino acids

in length (Liu et al., 2021). Moreover, these peptides are commonly combined with a carrier protein to facilitate their recognition and processing by antigen-presenting cells (APCs), thereby triggering an immune response that involves CTLs (Abd-Aziz & Poh, 2022).

To develop an effective peptide-based vaccine, it is crucial to ensure that the epitopes are recognizable by T cells, highly prevalent, and exclusive to tumor cells (Nelde et al., 2021). The process of identifying immunogenic epitopes begins by obtaining samples of both cancerous and normal cells through biopsy, followed by comparing their DNA sequences using techniques like Whole Exome Sequencing (WES) or Whole Genome Sequencing (WGS) to identify tumor-specific mutations (Richard et al., 2022). Somatic variant callers can be employed to detect single nucleotide variants from the WES and WGS data, and subsequently, peptides containing the mutated regions are extracted using sliding windows from the varied protein sequence (Richters et al., 2019). Epitope candidates can be identified using computational prediction tools, an approach that offers significant advantages in terms of time and cost when compared to traditional methods such as mass spectrometry techniques (Lemmel & Stevanović, 2003; Chen et al., 2020; Hensen et al., 2022; Parvizpour et al., 2020).

Immunogenicity refers to the capacity of a substance to initiate an immune response, and the binding of a peptide to HLA molecules is essential for epitope immunogenicity (Antunes et al., 2018). (Wu et al., 2019) devised DeepHLApan, a recurrent neural network-based method that integrates binding affinity and immunogenicity details of peptide-HLA complexes to predict CD8+ T-cell epitopes. (Li et al., 2021), on the other hand, utilized a convolutional neural network (CNN)-based method called DeepImmuno, where they utilized linear peptides of 9 to 10 amino acids and 4-digit class I HLA alleles as input to forecast the immunogenicity score of peptide-HLA pairs for T-cell immune responses. (Diao et al., 2022) introduced Seq2Neo, a CNN-based pipeline that leverages the binding affinity and transporters associated with antigen processing (TAP) transport efficiency of a given peptide-HLA pair to improve the prediction of immunogenicity scores for T-cell immune responses.

Generative Adversarial Networks (GANs) (Goodfellow et al., 2014) are deep learning models with a generative model trying to learn and capture the distribution of training data, thereby being capable of synthesizing samples from the learned distribution (Gonog & Zhou, 2019; Creswell et al., 2018). GAN algorithms have found application in generating novel protein and peptide structures for use in drug screening and the discovery stage (Lin et al., 2022). Dutta (2022) developed a GAN equipped with a graph CNN that predicts solubility and toxicity from molecular descriptors, guiding the generator network to produce new small molecules with desired drug properties. Putin et al. (2018) proposed a GAN architecture known as "RANC," which incorporates reinforcement learning to generate chemically diverse structures with desired features, with a focus on maintaining similar lengths of the SMILES string as their training data. Karimi et al. (2020) developed a semi-supervised, guided, conditional, Wasserstein generative adversarial network capable of generating proteins with desired structure folds, while incorporating greater sequence diversity and novelty compared to conditional variational auto-encoder designs. Li et al. (2021) employed a generator trained with a Wasserstein generative adversarial network with gradient penalty (WGAN-GP) to generate immunogenic epitopes binding to HLA-A*0201 and demonstrated that their generated peptides exhibited features and amino acid sequences similar to the epitopes in their training dataset.

In this study, we introduce a novel training framework, termed goal-directed Wasserstein Generative Adversarial Network with Gradient Penalty (GD-WGAN-GP), for generating immunogenic peptide sequences. Built upon the WGAN-GP architecture, known for its enhanced training stability (Gulrajani et al., 2017), our method integrates both a critic network and a reward network to guide the generator. The critic is trained on bladder cancer-specific peptides that bind to HLA-A*0201, assigning higher values to sequences similar to those in the training data. The reward network, implemented as a convolutional neural network and trained using immunogenicity scores predicted by DeepImmuno (Li et al., 2021), directs the generator toward producing peptides with high immunogenicity. Its output is scaled by a factor $S_{scale}$ to balance the influence of the reward and the critic in the loss of the generator. To reduce repetition among generated sequences, we apply two strategies: a switching mechanism that excludes the reward term when duplicated peptides are present in a batch, and otherwise multiplies it by a $\gamma_{max}$ parameter to control the reward's contribution to the loss of the generator; and a repetition penalty adapted from ORGAN (Guimaraes et al., 2018), where each reward is divided by the number of occurrences of its corresponding peptide within the

batch. The proposed GD-WGAN-GP with the switching mechanism enables users to control the trade-off between generating peptides with high predicted immunogenicity at the cost of increased repetition, or achieving lower repetition rates with reduced immunogenicity. In both cases, the generated peptides maintain strong binding affinity to HLA-A*0201, a characteristic shared with peptides in the training dataset. The generator trained using the GD-WGAN-GP framework with ORGAN's repetition penalty (Guimaraes et al., 2018) demonstrates the ability to produce peptides with high predicted immunogenicity, low repetition rates, and strong binding affinity to HLA-A*0201. These improvements outperform existing approaches, such as WGAN-GP (Li et al., 2021) and MolGAN (Cao & Kipf, 2022), highlight the potential of our method for the design of immunogenic epitope candidates in cancer immunotherapy. An introduction to our method is shown in Fig. 1.

## 2 Methods

### 2.1 Datasets

The tumor-specific neoantigen database (TSNAdb) (Wu et al., 2018) offers 6234 bladder cancer neoepitopes, each featuring a single amino acid mutation, predicted to bind with HLA-A*0201 and having predicted binding affinity $IC_{50}$ <500nM. These epitopes, along with their corresponding HLA molecules, were predicted using NetMHCpan v4.0 (Jurtz et al., 2017) which assesses the binding affinity of potential epitopes within a protein sequence based on mass spectrometry eluted ligands and half-maximal inhibition (IC50) scores, with a threshold of IC50 < 500nM. It is important to note that these epitopes have not been experimentally verified in other literature. We choose HLA-A*0201 as the binding target of the generated peptides since HLA-A*0201 binds with most of the bladder cancer neoepitopes in the TSNAdb (Wu et al., 2018).

During the data cleaning process, we specifically selected bladder cancer epitope sequences predicted to bind with HLA-A*0201 and having a length of either 9 or 10 amino acids. Our resulting training dataset comprised a total of 6234 epitopes. For standardization, we employed the same coding strategy as in (Li et al., 2021). This involved padding 9-mer peptides to become 10-mers by joining the first five amino acids and the last four amino acids with a placeholder "-". Thus, the peptide sequences in our study are represented as a sequence consisting of a placeholder "-" and the 20 amino acid types: Alanine (A), Arginine (R), Asparagine (N), Aspartate (D), Cysteine (C), Glutamine (Q), Glutamate (E), Glycine (G), Histidine (H), Isoleucine (I), Leucine (L), Lysine (K), Methionine (M), Phenylalanine (F), Proline (P), Serine (S), Threonine (T), Tryptophan (W), Tyrosine (Y), and Valine (V). The amino acids and placeholder were converted into one-hot encoded matrices (Jiang et al., 2022).

As of March 2023, the Immune Epitope Database (IEDB) (Vita et al., 2019) reports only 24 experimentally tested linear bladder cancer epitopes that bind with HLA class I molecules, as identified by Wang et al. (2020). In addition, the neoepitopes from the TSNAdb (Wu et al., 2018) are not predicted to be immunogenic. Thus, we aim to design a goal-directed generator to provide potential immunogenic peptide sequences and increase the pool of peptides worthy of experimental testing.

### 2.2 Goal-directed WGAN-GP

The designed GD-WGAN-GP architecture comprises a peptide sequence generator, a critic, an immunogenicity predictor, and a reward network. The generator and critic are CNN-based models from (Li et al., 2021). The critic is trained using the predicted bladder cancer peptides obtained from TSNAdb (Wu et al., 2018) with IC50 < 500nM and the generated epitopes, thereby guiding the training of the generator. Before being input to the critic, the predicted bladder cancer epitopes are encoded as a one-hot matrix, as depicted in Figure 4 (**a**) in Appendix A. Conversely, the output of the generator is a matrix of probabilities, which can be decoded as a peptide sequence, as shown in Figure 4 (**b**) in Appendix A. The peptide sequence generator tries to generate sequences that are closely similar to the predicted bladder cancer epitopes in the training dataset.

The immunogenicity predictor utilized is the Deepimmuno-CNN from (Li et al., 2021). This predictor takes a 9-mer or 10-mer generated peptide sequence, along with a 46-mer HLA-A*0201 sequence, as input,

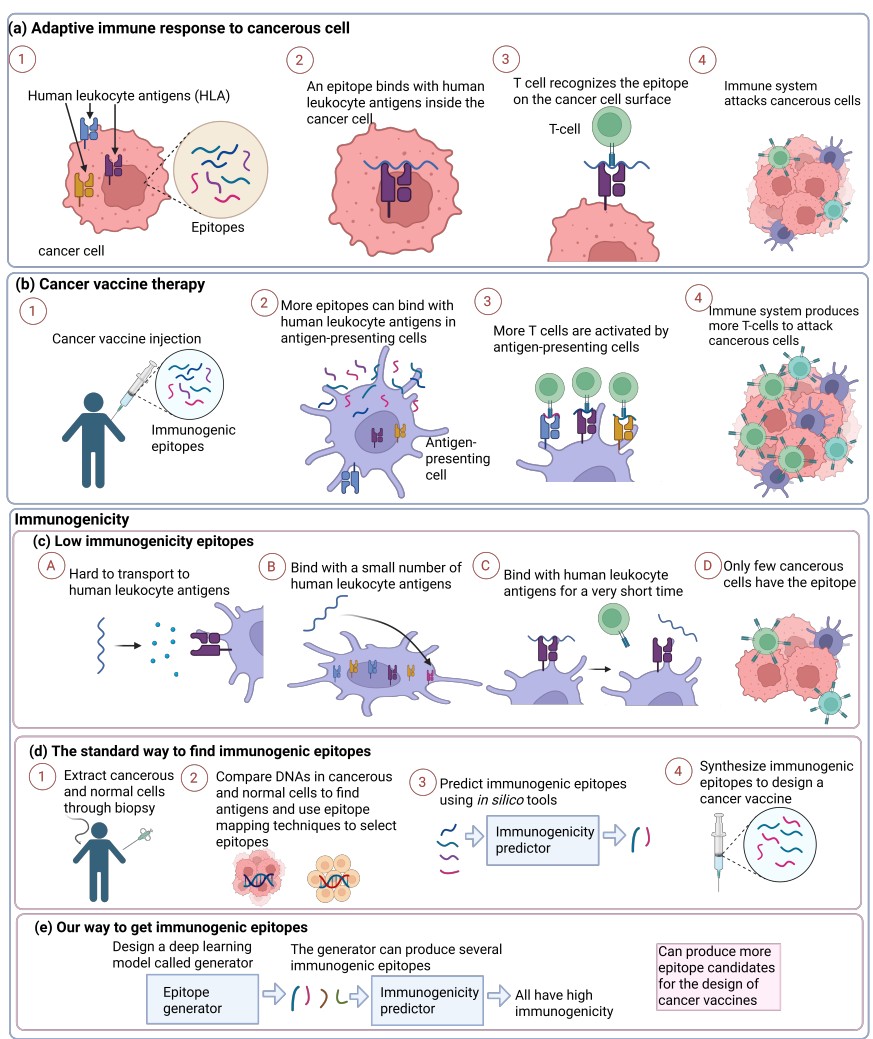

Figure 1: Introduction to peptide-based cancer vaccines and our proposed method. (a) Epitopes in cancerous cells may bind to HLA and the resulting HLA-peptide complex may be transferred to the cell surface to be recognized by T cells Zhao et al. (2021). (b) The polypeptides in peptide-based cancer vaccines are processed in APCs, the epitopes are bound to HLA, and the HLA-complexes present the epitopes on the cell surface to trigger T cell immune responses Bartnik et al. (2013). (c) The immunogenicity of a peptide can be influenced by its delivery system Li et al. (2014), the binding diversity between peptide and HLA Stephens et al. (2021), the stability of the HLA-peptide complex Van Der Burg et al. (1996), and the abundance and density of the antigen that is presented on a cell surface Purcell et al. (2007). (d) The standard way of finding the immunogenic peptides for the design of cancer vaccines Guo et al. (2018). (e) The designed generator can produce more epitope candidates with high immunogenicity to facilitate peptide-based cancer vaccines. Created with BioRender.com.

predicting their respective immunogenicities. The reward network is a CNN-based model that outputs the immunogenicity score. In contrast to the immunogenicity predictor, the reward network is trained during the training of the GD-WGAN-GP. The reward network takes the generated sequences or the sequences in the training dataset as input and the output of the immunogenicity predictor given the reward network's input as the target value. Throughout the training, the placeholder '-' remains in the peptide sequence and is only removed when utilizing the trained generator to produce peptide sequences, as depicted in the procedure outlined in Figure 4 (**b**), step **(3)** in Appendix A.

Consider a generative network $G_\theta : \mathbb{R}^m \to \mathbb{R}^{p \times q}$ with a set of parameters $\theta \in \mathbb{R}$ and a regression model $P : \mathbb{R}^{p \times q} \to [0, 1]$. We aim to update the set of parameters $\theta$ such that the expected value $E_{G_\theta(z) \sim \mathbb{P}_G}[\hat{p}]$ is maximized, where $\hat{p} = P(G_\theta(z))$, $z \in \mathbb{R}^m$ is a random noise vector, and $\mathbb{P}_G$ is the generated data distribution.

The loss function of the critic $D_\omega$ is the same as in (Gulrajani et al., 2017), which is

$$
\begin{aligned}
L_\omega = &E_{\tilde{x} \sim \mathbb{P}_G}[D_\omega(\tilde{x})] - E_{x \sim \mathbb{P}_r}[D_\omega(x)] \\
&+ \lambda E_{\hat{x} \sim \mathbb{P}_{\hat{x}}}[(\|\nabla_{\hat{x}} D_\omega(\hat{x})\|_2 - 1)^2],
\end{aligned}
\tag{1}
$$

where

$$
\tilde{x} = G_\theta(z), \tag{2}
$$

$$
\hat{x} = \epsilon x + (1 - \epsilon)\tilde{x}, \tag{3}
$$

$E_{x \sim \mathbb{P}}[f(x)]$ is the expected value of a function $f(x)$ with $x$ sampled from the distribution $\mathbb{P}$, $\mathbb{P}_r$ and $\mathbb{P}_G$ are the training data distribution and the generated data distribution, respectively, $\mathbb{P}_{\hat{x}}$ is the distribution of the data sampled from the training and generated distribution as defined in (3), $G_\theta(\cdot)$ and $D_\omega(\cdot)$ is the function of the generator and the critic in WGAN-GP, $\lambda$ is the penalty coefficient, $\|\cdot\|_2$ is the L-2 norm, $\epsilon \in [0, 1]$ is a random real number between zero and one, $z$ is a random noise vector and each element is sampled from a normal distribution with zero mean and unit variance, $\theta$ is the weights in the generative network $G_\theta$, and $\omega$ is the weights in the critic network $D_\omega$.

### 2.2.1 Generator loss with repetition penalty

**The first approach: switching between vanilla WGAN-GP and a combined loss with the output of the reward network**   The designed loss function of the generator $G_\theta$ with a switching mechanism is

$$
L_\theta^{D1} = -E_{\tilde{x} \sim \mathbb{P}_G}[D_\omega(\tilde{x})] - \gamma S_{scale} E_{\tilde{x} \sim \mathbb{P}_G}[S_\phi(\tilde{x})], \tag{4}
$$

where

$$
\gamma = \begin{cases} \gamma_{max} & \text{if } n(A) = N_{batch} \\ 0 & \text{otherwise} \end{cases}, \tag{5}
$$

$\gamma_{max} \in [0, 1]$, $A$ is a set composed of the generated one-probability matrix $G_\theta(z)$ in a batch, $n(A)$ is the cardinality of $A$ (the number of elements in the set $A$), $N_{batch}$ is the number of generated peptides in a batch,

$$
S_{scale} = 1 + |E_{\tilde{x} \sim \mathbb{P}_G}[D_\omega(\tilde{x})] - E_{\tilde{x} \sim \mathbb{P}_G}[S_\phi(\tilde{x})]|, \tag{6}
$$

$|\cdot|$ denotes as modulus, $S_\phi$ is the reward network, and $\phi$ is the weights in the reward network $S_\phi$.

The loss function of the reward network $S_\phi$ is defined as

$$
L_\phi = E_{\tilde{x} \sim \mathbb{P}_G}[(S_\phi(\tilde{x}) - P(\tilde{x}))^2] + E_{x \sim \mathbb{P}_r}[(S_\phi(x) - P(x))^2], \tag{7}
$$

where $P(\cdot) \in [0, 1]$ is the output of the immunogenicity predictor.

The variable $S_{scale}$ in the loss function of the generator (4) ensures that the expected value of the critic output will not dominate the generator's loss function. The variable $\gamma$ in the loss function of the generator (4) is used to force the generator to produce diverse peptide sequences in the training stage. If $\gamma = 1$, (4) becomes a weighted sum of the expected value of the critic output and the expected value of the reward network output given the generated one-probability matrices $G_\theta(z)$ as input. If $\gamma = 0$, the training of the GD-WGAN-GP is the vanilla WGAN-GP in Gulrajani et al. (2017). The architecture of GD-WGAN-GP using the first generator loss design is illustrated in Figure 2.

**The second approach: dividing each reward in the batch by its number of occurrences (Guimaraes et al., 2018)**   The loss function of the generator $G_\theta$ with the repetition penalty from ORGAN (Guimaraes et al., 2018) is

$$
L_\theta^{D2} = -E_{\tilde{x} \sim \mathbb{P}_G}[D_\omega(\tilde{x})] - S_{scale} E_{\tilde{x} \sim \mathbb{P}_G}[\hat{S}_\phi(\tilde{x})], \tag{8}
$$

where $\hat{S}_\phi(\tilde{x})$ is the output of the reward network, $S_\phi(\tilde{x})$, divided by the number of the occurrences of the generated one-hot matrix, $\tilde{x}$, in a batch.

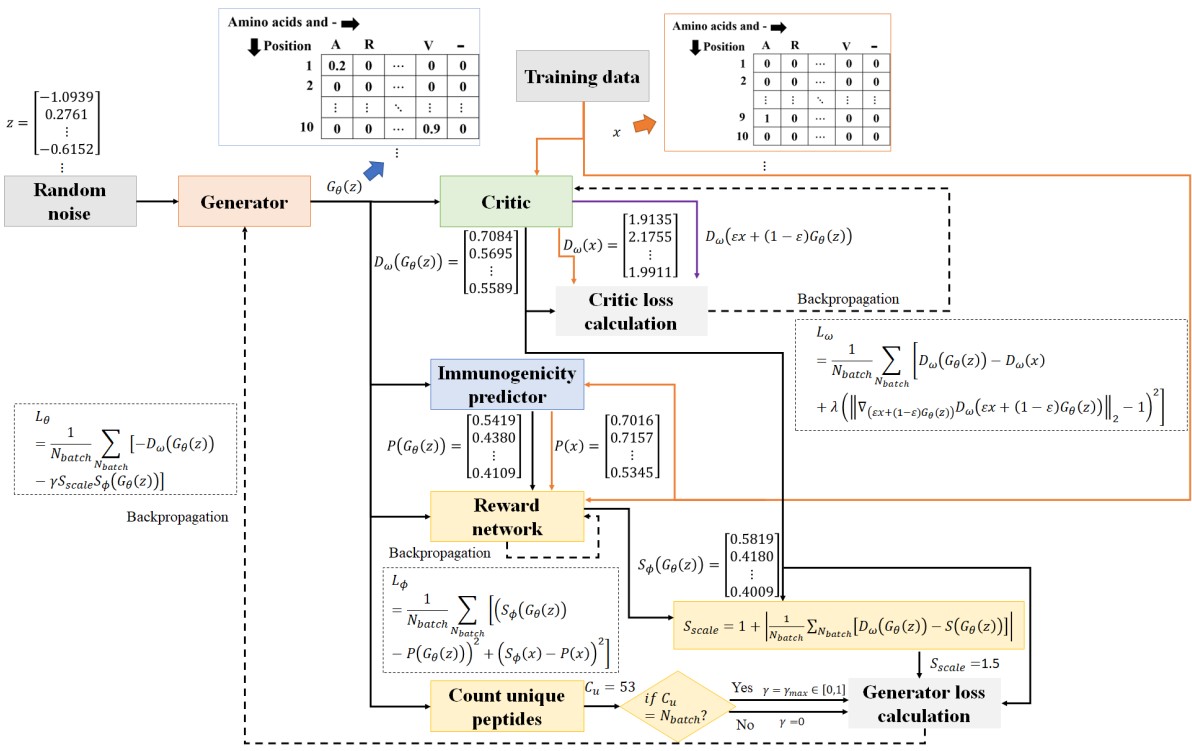

Figure 2: Training scheme of the designed GD-WGAN-GP with the switching mechanism in (4). The training of the GD-WGAN-GP involves incorporating an immunogenicity predictor from Li et al. (2021) to train a generator and a reward network. The objective of the reward network is to assist the generator in producing highly immunogenic peptides, while the critic aims to guide the generator in generating peptides with high similarity (though not identical) to those in the training dataset. The generator takes a set of $N_{batch}$ random noise vectors $z$, each composed of 128 elements sampled from a zero-mean, unit-variance Gaussian distribution, as input. Implemented as a deep convolutional neural network, the generator produces $N_{batch}$ one-probability matrices $G_\theta(z)$, which serve as input for the critic, immunogenicity predictor, and reward network. The critic's output is a vector of real values $D_\omega(G_\theta(z))$, the output of the immunogenicity predictor $P(G_\theta(z))$ is a vector of real values between 0 and 1, and the reward network's output is a vector with real values $S_\phi(G_\theta(z))$. The reward network is trained using the mean squared error between the output of the immunogenicity predictor and its own output, considering the generated one-probability matrices $G_\theta(z)$ and the one-hot encoded matrix of peptides in the training data $x$ as input. The outputs of the reward network $S_\phi(G_\theta(z))$ and the output of the critic $D_\omega(G_\theta(z))$ are used to compute a scaling variable $S_{scale}$. The variable $\gamma$ indicates whether the generated peptides are repeated in a batch ($\gamma = 0$ for repetitions, $\gamma = \gamma_{max}$ otherwise). The critic is trained using the standard critic loss in a WGAN-GP, while the generator is trained with a loss aiming to minimize the critic's output given the generated one-probability matrices $D_\omega(G_\theta(z))$ and to minimize the reward network's output given the generated one-probability matrices $S_\phi(G_\theta(z))$ multiplied by the scaling variable $S_{scale}$ and the variable $\gamma$.

### 2.2.2 Training of the GD-WGAN-GP

In this paper, the batch size is 64 ($N_{batch} = 64$). $z \in \mathbb{R}^{128}$ is defined to be a random noise vector with 128 elements, and each element is sampled from a normal distribution with zero mean and unit variance. The generative network, the critic network, and the reward network network are defined as $G_\theta : \mathbb{R}^{128} \to \mathbb{R}^{10 \times 21}$, $D_\omega : \mathbb{R}^{10 \times 21} \to \mathbb{R}$, and $S_\phi : \mathbb{R}^{10 \times 21} \to \mathbb{R}$, respectively. For each epoch, the critic is trained 97 times, the generator is trained 10 times, and the reward network is trained 97 times. The architecture of the

---

**Algorithm 1** The training procedure of the GD-WGAN-GP with the switching mechanism. The value for the parameters are $k = 1$, $\lambda = 10$, $n_{critic} = 10$, $N_{data} = 6232$, $N_{batch} = 64$, $\alpha = 10^{-4}$, $\beta_1 = 0.5$, and $\beta_1 = 0.9$.

> **Require:** A maximum iteration number $N$, $\gamma_{max}$, and the initial weights $\omega_0$, $\theta_0$, and $\phi_0$ for the critic $D_\omega$, the generator $G_\theta$, and the reward network $S_\phi$, respectively.
> $n_{split} \leftarrow \lfloor \frac{N_{data}}{N_{batch}} \rfloor$
> **while** $k \leq N$ **do**
>> Randomly split the training dataset with data distribution $\mathbb{P}_r$ to $n_{split}$ data $\{\mathbb{P}_r^{(1)}, \mathbb{P}_r^{(2)}, \cdots, \mathbb{P}_r^{(n_{split})}\}$
>> **for** $i = 1, \cdots, n_{split}$ **do**
>>> Sample a batch of noise vectors $\{z^{(j)}\}_{j=1}^{N_{batch}} \sim N(0,1)$.
>>> **for** $j = 1, \cdots, N_{batch}$ **do**
>>>> Sample real data $x \sim \mathbb{P}_r^{(i)}$ without replacement, and a random number $\epsilon \in U[0,1]$.
>>>> $\tilde{x} \leftarrow G_\theta(z^{(j)})$
>>>> $L_\phi^{(j)} \leftarrow \frac{1}{N_{batch}}((S_\phi(\tilde{x}) - P(\tilde{x}))^2 + (S_\phi(x) - P(x))^2)$
>>>> $\hat{x} \leftarrow \epsilon x + (1 - \epsilon)\tilde{x}$
>>>> $L_\omega^{(j)} \leftarrow \frac{1}{N_{batch}}(-D_\omega(x) + D_\omega(\tilde{x}) + \lambda(\|\nabla_{\hat{x}} D_\omega(\hat{x})\|_2 - 1)^2)$
>>> **end for**
>>> $\phi \leftarrow \text{Adam}(\nabla_\phi \sum_{j=1}^{N_{batch}} L_\phi^{(j)}, \phi, \alpha, \beta_1, \beta_2)$
>>> $\omega \leftarrow \text{Adam}(\nabla_\omega \sum_{j=1}^{N_{batch}} L_\omega^{(j)}, \omega, \alpha, \beta_1, \beta_2)$
>>> **if** $(i \mod n_{critic}) = 0$ **then**
>>>> Sample a batch of noise vectors $\{z^{(j)}\}_{j=1}^{N_{batch}} \sim N(0,1)$.
>>>> $A \leftarrow \emptyset$
>>>> **for** $j = 1, \cdots, N_{batch}$ **do**
>>>>> Add $G_\theta(z^{(j)})$ to a set $A$
>>>> **end for**
>>>> $S_{scale} \leftarrow 1 + |\frac{1}{N_{batch}} \sum_{j=1}^{N_{batch}} D_\omega(G_\theta(z^{(j)})) - \frac{1}{N_{batch}} \sum_{j=1}^{N_{batch}} S_\phi(G_\theta(z^{(j)}))|$
>>>> **if** $n(A) = N_{batch}$ **then**
>>>>> $\gamma \leftarrow \gamma_{max}$
>>>> **else**
>>>>> $\gamma \leftarrow 0$
>>>> **end if**
>>>> $L_\theta \leftarrow \frac{1}{N_{batch}} \sum_{j=1}^{N_{batch}} (-D_\omega(G_\theta(z^{(j)})) - \gamma S_{scale} S_\phi(G_\theta(z^{(j)})))$
>>>> $\theta \leftarrow \text{Adam}(\nabla_\theta L_\theta, \theta, \alpha, \beta_1, \beta_2)$
>>> **end if**
>> **end for**
>> $k := k + 1$
> **end while**

---

generator, the critic, the reward network, and the residual block in these three networks can be found in Table 3, Table 4, Table 5, and Table 6 in Appendix A, respectively. The algorithm of the GD-WGAN-GP with switching mechanism in (4) is shown in Algorithm 1. The same hyperparameters are used for the GD-WGAN-GP with the ORGAN's repetition penalty (Guimaraes et al., 2018) in (8).

## 3 Experiments

To validate the efficacy of our proposed generator training scheme, we adopted the same architecture for both the generative and critic networks as outlined in (Li et al., 2021). We then compared the peptides generated by the generator trained with and without our devised training scheme. For assessing immunogenicity, we utilized the CNN-based immunogenicity predictor, DeepImmunoCNN, from (Li et al., 2021). The CNN was retrained using the "remove0123_sample100.csv" data file provided in their source code. The immunogenicity predictor takes a 9- or 10-mer peptide sequence and a 4-digit encoded HLA sequence as inputs, predicting their real-value immunogenicity score within the range of $[0, 1]$.

For evaluation, we performed a single run in which each generator produced 10,000 peptide sequences using the same noise matrix. The matrix consists of 10,000 rows, each representing a batch, and 128 columns, each corresponding to a dimension of the noise vector. The same set of generated sequences was used for both immunogenicity and binding affinity evaluations.

## 3.1 Compared method

The designed GD-WGAN-GP is compared with the MolGAN designed by Cao & Kipf (2022). Cao & Kipf (2022) used WGAN without gradient penalty, and the loss functions of their critic and generator can be written as:

$$L_\omega^{MolGAN} = E_{\tilde{x} \sim \mathbb{P}_G}[D_\omega(\tilde{x})] - E_{x \sim \mathbb{P}_r}[D_\omega(x)] \tag{9}$$

and

$$L_\theta^{MolGAN} = -\lambda_M E_{\tilde{x} \sim \mathbb{P}_G}[D_\omega(\tilde{x})] - (1 - \lambda_M)S_\phi(\tilde{x}), \tag{10}$$

respectively, where $\lambda_M \in [0, 1]$. The same hyperparameters as the designed GD-WGAN-GP are used for the training of the MolGAN (Cao & Kipf, 2022) and the weight clipping of $c = 0.01$ is applied to the critic network.

## 3.2 Immunogenicity

A comparison between the generators trained with the designed GD-WGAN-GP training scheme and the generator trained using the architecture from Li et al. (2021) and MolGAN (Cao & Kipf, 2022) is shown in Table 1. The GANs are trained on bladder cancer epitope data from TSNAdb (Wu et al., 2018). Five variants of the proposed GD-WGAN-GP are evaluated. The first four variants apply the switching mechanism in (4) with $\gamma_{max}$ set to 0.25, 0.5, 0.75, and 1, respectively. The fifth variant incorporates the repetition penalty from ORGAN (Guimaraes et al., 2018), as defined in (8). All of them are trained after 1000 epochs.

Three variants of the MolGAN (Cao & Kipf, 2022) designs are compared with the proposed GD-WGAN-GP. The first two variants use MolGAN with $\lambda_M$ set to 0.5 and 0 in (10), respectively. The third variant uses MolGAN with $\lambda_M = 0$ and incorporates the repetition penalty from ORGAN (Guimaraes et al., 2018). For each variant, two generator checkpoints are selected: one after 1000 training epochs, and another selected based on the epoch that yields the highest sum of the immunogenicity score and the ratio of non-repeated peptides in a batch of 64 generated samples across all 1000 epochs, following a model selection strategy similar to that used in Cao & Kipf (2022).

In Table 1, 10,000 peptide sequences are produced by each of the generator given the same input with a size of $\mathbb{R}^{10000 \times 128}$, where each element is sampled from a normal distribution. The percentage of non-repeated peptides for each generator is shown in the fourth column of Table 1. The five variants of the proposed GD-WGAN-GP achieve percentages above 75%, with one reaching 99.98%, outperforming the MolGAN (Cao & Kipf, 2022) designs, which remain below 25%.

To ensure that the generated peptides can be used as the input for the immunogenicity predictor (Li et al., 2021), the peptide sequences containing more than 2 placeholders ('-') are removed. In addition, the repeated peptide sequences are excluded prior to evaluating the immunogenicity score. The average immunogenicity score across the remaining peptides for each generator is presented in the second column in Table 1.

The GD-WGAN-GP with the switching machenism allows different generator configurations to optimize the predicted immunogenicity score or the number of non-repeated peptides. The GD-WGAN-GP with ORGAN's (Guimaraes et al., 2018) repetition penalty can achieve a higher predicted immunogenicity score with a similar percentage of non-repeated peptides compared to GD-WGAN-GP with ($\gamma_{max} = 0.5$).

Using a model selection strategy similar to that used in MolGAN (Cao & Kipf, 2022), which selects the generator that maximizes the sum of the predicted immunogenicity score and the ratio of non-repeated peptides, the GD-WGAN-GP with ORGAN's (Guimaraes et al., 2018) repetition penalty achieves the highest sum of 1.763. The second-highest value, 1.7269, is obtained by the GD-WGAN-GP with $\gamma_{max} = 0.5$.

Details of the computational times are provided and discussed in Table 7 in Appendix B.2, and the dots and box plots of the immunogenicity scores across different methods, including the methods tested in this

Table 1: Comparison of different designs of GANs trained to generate peptide vaccine candidates for bladder cancer for 1000 epochs except for MolGAN[best] (Cao & Kipf, 2022), where the generator checkpoint is selected from the epoch that yields the highest sum of the immunogenicity score (imm. score) and the ratio of non-repeated peptides among a batch of 64 generated samples across all 1000 epochs. The imm. score is predicted by the predictor from Li et al. (2021) and its average is across the generated non-repeated peptides after removing peptides more than 10-mer and less than 9-mer. The percentage (percent.) of non-repeated peptides is the percentage of the value of the number of non-repeated peptides divided by the number of generated peptides (10,000). "min." indicates minute. In the Algorithm column, "with ORGAN" indicates the reward for each generated data is penalized by divided the number of its repetition in a batch during training.

| Algorithm | Average imm. score | Percent. of peptides with 9-10-mer (%) | Percent. of non-repeated peptides (%) | Total training time (min.) |
|---|---|---|---|---|
| WGAN-GP | 0.58±0.12 | 98.86 | 99.99 | 33.74 |
| MolGAN ($\lambda_M = 0.5$) | 0.72±0.08 | 63.28 | 18.13 | 181.14 |
| MolGAN[best] ($\lambda_M = 0.5$) | 0.73±0.09 | 65.46 | 17.44 | 166.75 |
| MolGAN ($\lambda_M = 0$) | 0.89±0.00 | 100.00 | 0.01 | 208.58 |
| MolGAN[best] ($\lambda_M = 0$) | 0.73±0.08 | 100.00 | 17.15 | 200.27 |
| MolGAN ($\lambda_M = 0$) with ORGAN | 0.89±0.00 | 100.00 | 0.01 | 203.77 |
| MolGAN[best] ($\lambda_M = 0$) with ORGAN | 0.73±0.08 | 100.00 | 16.81 | 160.56 |
| GD-WGAN-GP ($\gamma_{max} = 0.25$) | 0.63±0.11 | 99.86 | **99.98** | 218.16 |
| GD-WGAN-GP ($\gamma_{max} = 0.5$) | 0.73±0.10 | 99.50 | 99.69 | 216.58 |
| GD-WGAN-GP ($\gamma_{max} = 0.75$) | 0.88±0.05 | 99.95 | 81.60 | 216.49 |
| GD-WGAN-GP ($\gamma_{max} = 1$) | **0.92**±0.04 | 99.96 | 77.55 | 182.35 |
| GD-WGAN-GP with ORGAN | 0.77±0.10 | 99.93 | 99.30 | 193.89 |

section, the ablated variants of GD-WGAN-GP, the variants with different generator architectures, and the predictions from the IEDB predictor Vita et al. (2019), are shown in Figure 5 in Appendix B. The diversity of the generated peptides are discussed in Figure 8 of Appendix B.6

### 3.2.1 Evaluation of the immunogenicity on the generated brain cancer epitopes

The same evaluation is conducted on the same set of GANs trained using brain cancer epitopes from TSNAdb (Wu et al., 2018), with results presented in Table 8 in Appendix C. Unlike the result in Table 1, the MolGAN[best] variant with $\lambda_M = 0.5$ achieves a higher percentage of non-repeated peptides compared to GD-WGAN-GP ($\gamma_{max} = 1$), but yields a lower predicted immunogenicity score of 0.68. Moreover, GD-WGAN-GP variants with $\gamma_{max} = 0.5$, $\gamma_{max} = 0.75$, and the ORGAN's repetition penalty outperform MolGAN[best] in both predicted immunogenicity score and the percentage of non-repeated peptides.

The remaining five MolGAN (Cao & Kipf, 2022) variants achieve a maximum of only 22.30% non-repeated peptides, whereas all GD-WGAN-GP variants maintain a minimum of 60.15%. In addition, the GD-WGAN-GP with ORGAN's (Guimaraes et al., 2018) repetition penalty achieves the highest sum of the predicted immunogenicity score and the ratio of non-repeated peptides with a value of 1.7917 compared to other methods. These results further demonstrate that the proposed GD-WGAN-GP outperforms MolGAN across different datasets.

The higher percentage of non-repeated peptides among the generated brain cancer epitopes compared to the generated bladder cancer epitopes may result from the lower median binding affinity of the brain dataset relative to the bladder dataset as shown in Figure 10 of Appendix B.

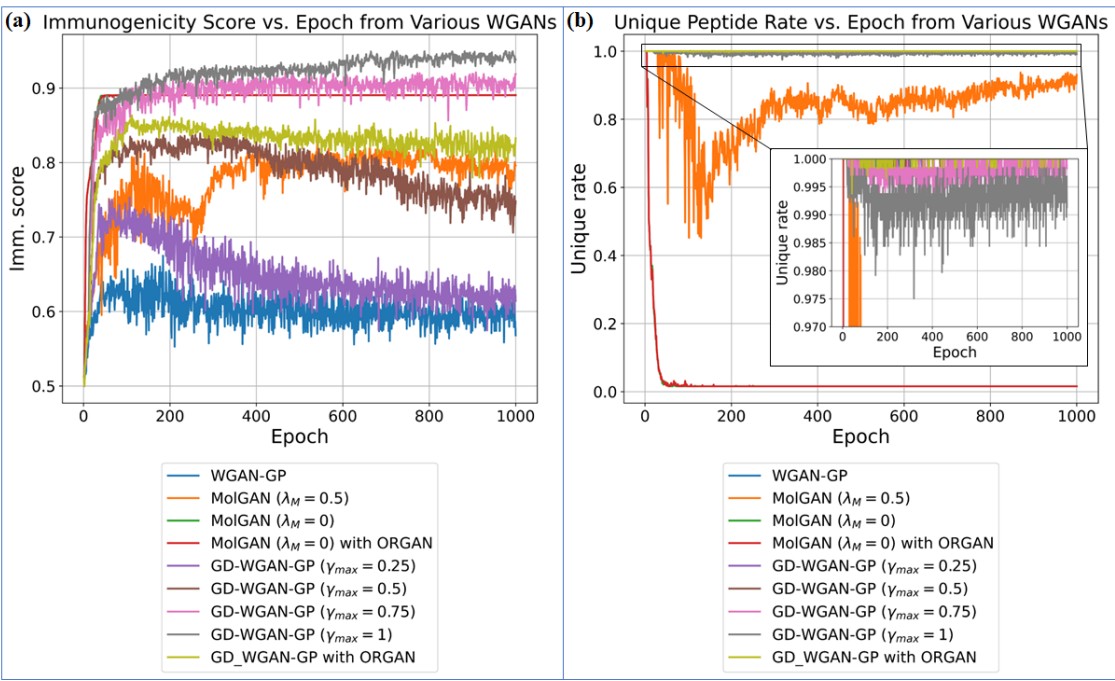

Figure 3: (**a**) The averaged immunogenicity score and (**b**) the unique rate of the peptide sequences generated from the proposed GD-WGAN-GPs (purple, brown, pink, gray, and apple green lines) compared to the WGAN-GP without the immunogenicity predictor (blue line) and MolGANs (Cao & Kipf, 2022) (orange, green, and red lines) through 1000 epochs of training. The proposed GD-WGAN-GP with $\gamma_{max} = 1$ achieves a higher immunogenicity and unique rate compared to MolGANs (Cao & Kipf, 2022) after 400 epochs.

### 3.2.2 Immunogenicity score and unique peptide rate during training

The average immunogenicity scores of the generated peptide sequences from the same set of generators in Section 3.2 during the 1000 epochs of training are presented in Figure 3 (a), where the solid lines represent the averaged scores calculated using

$$P_{epoch}^{(s)} = \frac{1}{10} \sum_{k=1}^{10} P_{mean}^{(k)}, \tag{11}$$

where $k = 1, 2, \cdots, 10$ is the number of times the generator is trained in an epoch, $s = 1, 2, \cdots, N$ is the number of epochs, $N = 1000$ is the maximum iteration number, and

$$P_{mean}^{(k)} = \frac{1}{N_{batch}} \sum_{j=1}^{N_{batch}} P(G_\theta(z^{(j)})), \tag{12}$$

Notably, the immunogenicity scores of sequences generated from the proposed GD-WGAN-GP continue to increase as the training epoch increases, reaching approximately 0.93 with the rate of non-repeated peptides (unique rate) of around 0.99 at epoch 1000, as shown in the gray line in Figure 3. Conversely, sequences generated without the immunogenicity predictor from WGAN-GP only reach around 0.6 and show no further increase. The MolGAN ($\lambda_M = 0$) and MolGAN ($\lambda_M = 0$) with ORGAN's repetition penalty achieve an immunogenicity scores of roughly 0.88, but the rate of non-repeated peptides decreases to around 0.02, as shown in the red and green lines in Figure 3 (**b**).

Table 2: Comparison of the predicted binding affinity between different designs of GANs trained to generate peptide vaccine candidates for bladder cancer after training for 1000 epochs except for MolGAN[best] (Cao & Kipf, 2022), where the generator checkpoint is selected from the epoch that yields the highest sum of the immunogenicity score and the ratio of non-repeated peptides among a batch of 64 generated samples across all 1000 epochs. The percentage (percent.) of strong binders is the percentage of the ratio between the number of generated unique sequences having $IC_{50} < 500nM$ binding with HLA-A*0201 predicted by NetMHCpan v4.0 (Jurtz et al., 2017) and the number of generated peptides (10,000). In the Algorithm column, "with ORGAN" indicates the reward for each generated data is penalized by divided the number of its repetition in a batch during training.

| Algorithm | Percent. of unique binders (%) | Number of generated unique peptides with 9-10-mer | Number of generated peptides in the training set |
|---|---|---|---|
| WGAN-GP | 67.61 | 9907 | 0 |
| MolGAN ($\lambda_M = 0.5$) | 4.68 | 1163 | 0 |
| MolGAN[best] ($\lambda_M = 0.5$) | 3.43 | 1246 | 0 |
| MolGAN ($\lambda_M = 0$) | 0.00 | 1 | 0 |
| MolGAN[best] ($\lambda_M = 0$) | 0.00 | 1708 | 0 |
| MolGAN ($\lambda_M = 0$) with ORGAN | 0.00 | 1 | 0 |
| MolGAN[best] ($\lambda_M = 0$) with ORGAN | 0.00 | 1677 | 0 |
| GD-WGAN-GP ($\gamma_{max} = 0.25$) | 77.64 | 9986 | 0 |
| GD-WGAN-GP ($\gamma_{max} = 0.5$) | 78.48 | 9919 | 0 |
| GD-WGAN-GP ($\gamma_{max} = 0.75$) | 62.13 | 8139 | 0 |
| GD-WGAN-GP ($\gamma_{max} = 1$) | 50.78 | 7730 | 0 |
| GD-WGAN-GP with ORGAN | **90.51** | 9934 | 0 |

## 3.3 Binding affinity

The binding affinity between the generated peptides and HLA-A*0201 is quantified by the half-maximal inhibition ($IC_{50}$) values, representing the concentration of the test peptide resulting in 50% inhibition of the binding of a probe peptide (Jurewicz et al., 2019). In the training dataset, the bladder cancer epitopes are predicted to have a binding affinity smaller than 500nM using NetMHCpan v4.0 (Jurtz et al., 2017) by (Wu et al., 2018). To investigate whether the generated peptides also exhibit a strong binding affinity ($< 500$nM), we employed the same prediction tool NetMHCpan 4.0 (Jurtz et al., 2017) to predict the $IC_{50}$ values. Peptides with a predicted binding affinity $IC_{50}$ below 500nM are considered binders to HLA-A*0201 (Lundegaard et al., 2008).

The percentage of the number of binders of the generated peptides divided by the total number of generated peptides from the designed GD-WGAN-GP is presented in the second column of Table 2. The GD-WGAN-GP with ORGAN's repetition penalty shows that the generated peptides have a similar property to the training set (binders to HLA-A*0201) compared to WGAN-GP and . The variants from the designed GD-WGAN-GP with the switching mechanism achieve a higher binder rate with a minimum of 50.78% compared to MolGAN's variants with no more than 5%.

The prediction of the binding affinity between HLA-A*0201 and the generated brain cancer epitopes candidates from the same set of generators is presented in Table 9 of Appendix C. The results show that WGAN-GP achieves the highest percentage of unique binders at 93.79%, followed by GD-WGAN-GP with $\gamma_{max} = 0.25$ at 89.45%. The lowest percentage of unique binders produced by the proposed method is 51.89% (from GD-WGAN-GP with $\gamma_{max} = 1$), which is higher than the MolGAN variants (Cao & Kipf, 2022), where the best-performing model, MolGAN[best] ($\lambda_M = 0.5$), achieves only 14.72%.

The results from Table 2 and Table 9 suggest that increasing $\gamma_{\mathrm{max}}$ improves the average predicted immunogenicity score but reduces both the number of unique binders and the percentage of non-repeated peptides.

These findings indicate that the proposed GD-WGAN-GP achieves a higher percentage of unique binders than the MolGAN variants (Cao & Kipf, 2022), implying that the generated peptides are more similar to the training data, all of which are binders to HLA-A*0201 with $IC_{50} < 500$ nM predicted by NetMHCpan v4.0 (Jurtz et al., 2017).

## 4    Limitations

One potential limitation of this work is the risk of overfitting or reward hacking, since a single predictor is used for both training and evaluation. This setup may bias the generator toward exploiting the weaknesses of the predictor rather than capturing patterns for high immunogenicity sequences. Several strategies could help mitigate this issue. First, regularization terms such as L1- or L2-norm penalties could be introduced into the optimization objective. Second, a separate validation dataset could be employed to enable early stopping if the validation loss begins to increase during training. Third, multiple predictors could be incorporated, with their outputs combined through a weighted sum, to reduce reliance on any single model and improve robustness.

Another limitation is observed in the results of the binding affinity between bladder and brain cancer. The lower percentage of binders among the generated bladder cancer peptides compared to the brain cancer peptides may be attributed to the lower median binding affinity of the brain dataset. A possible way to address this limitation is to incorporate a binding affinity predictor into our framework, similar to the integration of the immunogenicity predictor. However, this approach depends on the availability and reliability of suitable predictors.

## 5    Conclusion

This paper presents various designs of goal-directed Wasserstein Generative Adversarial Networks with Gradient Penalty (GD-WGAN-GP) for training generators capable of producing 9- to 10-mer peptide sequences with high predicted immunogenicity, low repetition rate, and strong binding affinity. The output of a reward network, which is concurrently trained to predict immunogenicity scores during the training of the generator network, is incorporated into the GD-WGAN-GP architecture by multiplying it with a scaling factor, $S_{scale}$, to prevent the generator from solely maximizing the critic output regardless of the reward signal. To reduce repetition, two approaches are proposed: (1) a switching mechanism in which the reward term is excluded in the generator loss when duplicated peptides are present in a batch, and otherwise multiplied by a $\gamma_{max}$ parameter to control the reward's contribution to the generator loss, and (2) a repetition penalty from OR-GAN, which divides each reward by the number of occurrences of its corresponding peptide within a batch. A strong binding affinity is achieved by using a training dataset comprising 6,234 bladder cancer epitope sequences with predicted binding affinity $IC_{50} < 500$ nM to HLA-A*0201.

The GD-WGAN-GP variant with the switching mechanism achieves the highest average immunogenicity score among all compared methods when $\gamma_{max} = 1$. The number of repeated peptides can be reduced by decreasing $\gamma_{max}$, although this comes at the cost of lower average immunogenicity. The GD-WGAN-GP variant using ORGAN's repetition penalty achieves the highest combined score of immunogenicity and uniqueness, suggesting it as the most balanced and effective design. All GD-WGAN-GP variants outperform existing goal-directed GANs in terms of the percentage of unique binders with predicted $IC_{50} < 500$ nM, indicating that the proposed models can generate peptides with properties similar to those in the training dataset while maximizing immunogenicity and minimizing redundancy.

**Broader Impact Statement**

We present novel algorithms that enable goal-directed generation within the Wasserstein generative adversarial network with gradient penalty framework, specifically used to design peptide sequences with enhanced predicted immunogenicity score for peptide vaccine development against bladder cancer. By prioritizing

candidate sequences with higher predicted immunogenicity, our method has the potential to reduce both the experimental cost and time required to identify peptides that are worth testing in laboratory and clinical settings. Furthermore, our approach demonstrates generalization to brain cancer, suggesting broader applicability across multiple cancer types. However, the reliability of the generated peptides is limited by the expressiveness and accuracy of the predictor models used during training and evaluation.

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

## A  Detailed description of the network architecture

We present the details of the generator, the critic, the reward network, and the residual block in Table 3, Table 4, Table 5, and Table 6, respectively. Before being input to the critic, the epitopes are encoded as a one-hot matrix, as depicted in Figure 4 (**a**). Conversely, the output of the generator is a matrix of probabilities, which can be decoded as a peptide sequence, as shown in Figure 4 (**b**).

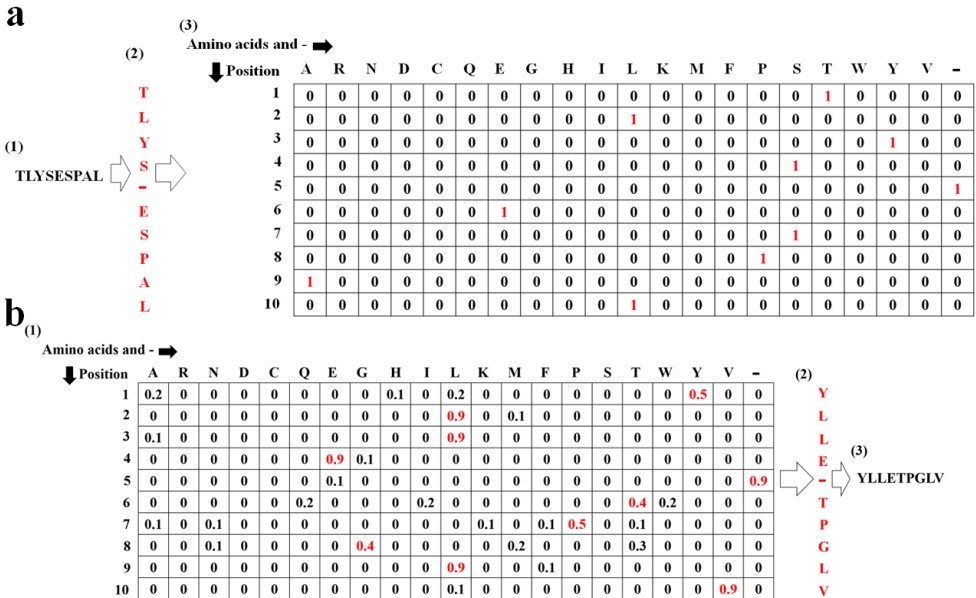

Figure 4: The encoding and the decoding methods in our goal-directed Wasserstein Generative Adversarial Network with Gradient Penalty (WGAN-GP) adopted from Li et al. (2021). In (**a**), the peptide is encoded from a peptide sequence to a one-hot matrix $x$. (**1**) A 9-mer peptide sequence, such as TLYSESPAL, is prepared before input to the critic. (**2**) If the peptide sequence has only 9 amino acids, a placeholder '-' is inserted at the fifth position to extend the sequence to a length of 10. (**3**) The encoded peptide is represented as a one-hot matrix, where each row corresponds to a position in the peptide sequence, and columns represent amino acids along with a placeholder '-'. The element in the one-hot matrix is 1 if the corresponding position contains the amino acid or the placeholder at the corresponding column. For example, 'T' is at position 1, so the first row of the one-hot matrix will have 1 at the 17th column, corresponding to the character 'T', and 0 in the other columns. In (**b**), the peptide is decoded through the generated one-probability matrix $G_\theta(z)$. (**1**) The output of the generator $G_\theta(z)$ is a one-probability matrix, where each row represents the position of a peptide sequence, and columns represent amino acids along with a placeholder '-'. Each element in $G_\theta(z)$ is the probability that the position corresponding to the row contains the amino acid or the placeholder. (**2**) A peptide sequence is decoded by selecting the amino acid or the placeholder with the highest probability for each row. For instance, if 0.5 is the highest value in the first row, the first character in the decoded sequence is 'Y', corresponding to the column with the value 0.5 in the first row. (**3**) If the peptide sequence contains a placeholder, it is removed to form a peptide sequence shorter than 10.

Table 3: Details of the generator network adopted from Li et al. (2021)

| Layer | Type | Kernel size | Filter | Stride | Padding | Output shape | Number of parameters |
|-------|------|-------------|--------|--------|---------|--------------|----------------------|
| 1 | Input | - | - | - | - | 128 | - |
| 2 | Fully Connected | - | - | - | - | 1280 | 165120 |
| 3 | Reshape | - | - | - | - | 128×10 | - |
| 4 | Residual block | - | - | - | - | 128×10 | 98560 |
| 5 | Residual block | - | - | - | - | 128×10 | 98560 |
| 6 | Residual block | - | - | - | - | 128×10 | 98560 |
| 7 | Residual block | - | - | - | - | 128×10 | 98560 |
| 8 | Residual block | - | - | - | - | 128×10 | 98560 |
| 9 | Convolution 1D | 1 | 21 | 1 | no | 21×10 | 2709 |
| 10 | Transpose | - | - | - | - | 10×21 | - |
| 11 | Gumbel-Softmax | - | - | - | - | 10×21 | - |

In the residual block shown in Table 6, the residual connection is defined to be

$$z_{res} = x_{res} + 0.3 y_{res}, \tag{13}$$

where $x_{res}$ is the input data in a residual block, $y_{res}$ is the output from the fifth layer (convolution 1D) in the residual block, and $z_{res}$ is the output of the residual connection.

We denote the output matrix of the Convolution 1D in the residual block as a two-dimensional matrix $Y^{res}$. The element at the $i$-th row and the $j$-th column of $Y^{res}$ is denoted $Y_{ij}^{res}$. The input matrix of the Convolution 1D in the residual block is denoted as $X^{res}$. In the residual block, the dimension of the input to the Convolution 1D is defined to be $X^{res} \in \mathbb{R}^{r_X^{res} \times c_X^{res}}$, where $r_X^{res} = 128$ and $c_X^{res} = 10$ are the number of rows and columns of the input matrix, respectively. The input matrix with one zero padding is defined as $X^{pad} = [\mathbf{0}^{res} \quad X^{res} \quad \mathbf{0}^{res}] \in \mathbb{R}^{r_X^{res} \times (c_X^{res}+2)}$, where $\mathbf{0}^{res} = [0 \quad 0 \quad \cdots \quad 0]^T \in \mathbb{R}^{r_X^{res} \times 1}$ is an all zero column vector with dimension $r_X^{res}$. The dimension of the output matrix after the Convolution 1D will be $Y^{res} \in \mathbb{R}^{f_{res} \times c_X^{res}}$, where $f_{res} = 128$ is the number of kernels. Each element in the output matrix of the Convolution 1D layer in the residual block is computed by

$$Y_{i,j}^{res} = \sum_{m=1}^{r_X^{res}} \sum_{n=1}^{c_w^{res}} (k_{m,n}^{res,i} \cdot X_{m,n+j}^{pad}) + b^{res,i}, \tag{14}$$

where $1 \leq i \leq f_{res}$, $1 \leq j \leq c_X^{res}$, $w^{res,i} \in \mathbb{R}^{r_X \times c_w^{res}}$ is the $i$-th convolutional kernel, $c_w^{res} = 3$ is the number of columns in the kernel, and $b^{res,i} \in \mathbb{R}$ is the bias for the $i$-th kernel.

For The Convolution 1D in the 9-th layer of the generator, $X^{gen} \in \mathbb{R}^{r_X^{gen} \times c_X^{gen}}$ is denoted as the input matrix, where $r_X^{gen} = 128$ and $c_X^{gen} = 10$ is the number of rows and columns of the input matrix, respectively. The output matrix will be $Y^{gen} \in \mathbb{R}^{f_{gen} \times c_X^{gen}}$, where $f_{gen} = 21$ is defined to be the number of kernels for this Convolution 1D. The Convolution 1D in the 9-th layer of the generator is defined to be

$$Y_{i,j}^{gen} = \sum_{m=1}^{r_X} \sum_{n=1}^{c_w^{gen}} (k_{m,n}^{gen,i} \cdot X_{m,n+j}^{gen}) + b^{gen,i}, \tag{15}$$

where $1 \leq i \leq f_{gen}$, $1 \leq j \leq c_X^{gen}$, $w^{gen,i} \in \mathbb{R}^{r_X \times c_w^{gen}}$ is the $i$-th convolutional kernel, $c_w^{gen} = 1$ is the number of columns in the kernel, and $b^{gen,i} \in \mathbb{R}$ is the bias for the $i$-th kernel.

For the Convolution 1D in the 3-rd layer of the critic and the reward network, $X^{critic} \in \mathbb{R}^{r_X^{critic} \times c_X^{critic}}$ is denoted as its input matrix, where $r_X^{critic} = 21$ and $c_X^{critic} = 10$ is the number of rows and columns of its input matrix, respectively. The output matrix will be $Y^{critic} \in \mathbb{R}^{f_{critic} \times c_X^{critic}}$, where $f_{critic} = 128$ is defined to be the number of kernels for this Convolution 1D. The operation of the Convolution 1D in the 3-rd layer

Table 4: Details of the critic network adopted from Li et al. (2021)

| Layer | Type | Kernel size | Filter | Stride | Padding | Output shape | Number of parameters |
|---|---|---|---|---|---|---|---|
| 1 | Input | - | - | - | - | 10×21 | - |
| 2 | Transpose | - | - | - | - | 21×10 | - |
| 3 | Convolution 1D | 1 | 128 | 1 | no | 128×10 | 2816 |
| 4 | Residual block | - | - | - | - | 128×10 | 98560 |
| 5 | Residual block | - | - | - | - | 128×10 | 98560 |
| 6 | Residual block | - | - | - | - | 128×10 | 98560 |
| 7 | Residual block | - | - | - | - | 128×10 | 98560 |
| 8 | Residual block | - | - | - | - | 128×10 | 98560 |
| 9 | Reshape | - | - | - | - | 1280 | - |
| 10 | Fully Connected | - | - | - | - | 1 | 1281 |

Table 5: Details of the reward network network

| Layer | Type | Kernel size | Filter | Stride | Padding | Output shape | Number of parameters |
|---|---|---|---|---|---|---|---|
| 1 | Input | - | - | - | - | 10×21 | - |
| 2 | Transpose | - | - | - | - | 21×10 | - |
| 3 | Convolution 1D | 1 | 128 | 1 | no | 128×10 | 2816 |
| 4 | Residual block | - | - | - | - | 128×10 | 98560 |
| 5 | Residual block | - | - | - | - | 128×10 | 98560 |
| 6 | Residual block | - | - | - | - | 128×10 | 98560 |
| 7 | Residual block | - | - | - | - | 128×10 | 98560 |
| 8 | Residual block | - | - | - | - | 128×10 | 98560 |
| 9 | Reshape | - | - | - | - | 1280 | - |
| 10 | Fully Connected | - | - | - | - | 1 | 1281 |

of the critic and the reward network is similar to (15) but with the change of the input size and the number of kernels and it is defined as

$$Y_{i,j}^{critic} = \sum_{m=1}^{r_X} \sum_{n=1}^{c_w^{critic}} (k_{m,n}^{dis,i} \cdot X_{m,n+j}^{critic}) + b^{dis,i}, \tag{16}$$

where $1 \le i \le f_{critic}$, $1 \le j \le c_X^{critic}$, $w^{dis,i} \in \mathbb{R}^{r_X \times c_w^{critic}}$ is the $i$-th convolutional kernel, $c_w^{critic} = 1$ is the number of columns in the kernel, and $b^{dis,i} \in \mathbb{R}$ is the bias for the $i$-th kernel.

In the critic and the reward network, the reshape function aligns each row into a row vector. The reshape function acted on the matrix $A \in \mathbb{R}^{p \times q}$ can be represented by

$$vec(A) \triangleq \begin{bmatrix} A_{r1} & A_{r2} & \cdots & A_{rp} \end{bmatrix}, \tag{17}$$

where $A_{ri}$ is denoted as the $i$-th row vector in the matrix $A$.

In the generator, the reshape function at the third layer converts a vector $B \in \mathbb{R}^{p \cdot q}$ into a matrix $\hat{B} \in \mathbb{R}^{p \times q}$ by putting the $p$-th $q$ elements of $B$ to the $p$-th row of the matrix $\hat{B}$. It can be represented as

$$\hat{B} = \begin{bmatrix} B_1 & B_2 & \cdots & B_q \\ B_{q+1} & B_{q+2} & \cdots & B_{2q} \\ \vdots & & & \\ B_{(p-1)q+1} & B_{(p-1)q+2} & \cdots & B_{pq} \end{bmatrix}, \tag{18}$$

Table 6: Details of the residual block adopted from Li et al. (2021)

| Layer | Type | Kernel size | Filter | Stride | Padding | Output shape | Number of parameters |
|---|---|---|---|---|---|---|---|
| 1 | Input | - | - | - | - | 128×10 | - |
| 2 | ReLU | - | - | - | - | 128×10 | - |
| 3 | Convolution 1D | 3 | 128 | 1 | 1 | 128×10 | 49280 |
| 4 | ReLU | - | - | - | - | 128×10 | - |
| 5 | Convolution 1D | 3 | 128 | 1 | 1 | 128×10 | 49280 |
| 6 | Residual connection | - | - | - | - | 128×10 | - |

where $B_i$ is denoted as the $i$-th elements in the matrix $B$.

Let $X^{GS} \in \mathbb{R}^{r_X^{GS} \times c_X^{GS}}$ be the input and $Y^{GS} \in \mathbb{R}^{r_X^{GS} \times c_X^{GS}}$ be the output of the Gumbel-Softmax in the 11-th layer of the generator network, where $r_X^{GS} = 10$ is the number of row for the input matrix $X^{GS}$ and $c_X^{GS} = 21$ is the number of column in $X^{GS}$, respectively. The Gumbel-Softmax in the generator network is computed as

$$Y_{i,j}^{GS} = \frac{e^{X_{i,j}^{Gumbel}}}{\sum_{m=1}^{c_X^{GS}} e^{X_{i,m}^{Gumbel}}}, \tag{19}$$

where $1 \le i \le r_X^{GS}$, $1 \le j \le c_X^{GS}$, $Y_{i,j}^{GS}$ is denoted as the element of $Y^{GS}$ at the $i$-th row and the $j$-th column, $X^{Gumbel}$ is evaluated as

$$X^{Gumbel} = \frac{X^{GS} - ln(g^{GS})}{\tau}, \tag{20}$$

where $g^{GS} \in \mathbb{R}^{r_X^{GS} \times c_X^{GS}}$ is a matrix with its value generated by the exponential distribution $f(x) = \lambda e^{-\lambda x}$, $\lambda = 1$, and $\tau = 0.75$.

## B  Additional experiment comparison for bladder cancer epitopes generation

### B.1  Additional baselines

A random generator is included as a baseline method. It is implemented by first uniformly sampling either 9 or 10 to determine the sequence length, and then uniformly sampling one character from "ARNDC-QEGHILKMFPSTWYV" for each position in the sequence.

### B.2  Additional comparison of computational time

The detail of the computational time for each process during the 1000 training for each method is shown in Table 7, showing that the main reason for the increased computational time for all the other methods compared to WGAN-GP resulting from the computation of the forward pass of the immunogenecity predictor.

### B.3  Additional comparison of immunogenecity score and binding affinity

The dots and box plots of the immunogenicity score from the Deepimmuno-CNN Li et al. (2021) (used in the training) and IEDB Vita et al. (2019) (unseen) across different methods are shown in Figure 5. The results from the IEDB predictor show that the GD-WGAN-GP variants with CNN generators achieve higher immunogenicity scores compared to the random generator baseline, the training dataset, and WGAN-GP. In addition, the MolGAN variants ($\lambda_M = 0$) obtain higher scores from the IEDB predictor Vita et al. (2019) than other methods, which is not observed in the results from DeepImmuno-CNN Li et al. (2021).

The dots and box plots of the immunogenicity scores predicted by NetMHCpan-4.0 Jurtz et al. (2017) and NetMHCpan-4.1 Reynisson et al. (2020) are shown in Figure 6. The results indicate that the predictions from

Table 7: Comparison of computational time of different designs of GANs trained to generate peptide vaccine candidates for bladder cancer for 1000 epochs except for MolGAN[best] (Cao & Kipf, 2022), where the generator checkpoint is selected from the epoch that yields the highest sum of the immunogenicity score (imm. score) and the ratio of non-repeated peptides among a batch of 64 generated samples across all 1000 epochs. "min." indicates minute. In the Algorithm column, "with ORGAN" indicates the reward for each generated data is penalized by divided the number of its repetition in a batch during training.

| Algorithm | Weight update time (min.) | | | Forward pass time (min.) | | | | Others (min.) | Total training time (min.) |
|---|---|---|---|---|---|---|---|---|---|
| | Critic | Generator | Reward | Critic | Generator | Reward | Predictor | | |
| WGAN-GP | 13.47 | 0.71 | - | 3.71 | 2.36 | - | 7.32 | 6.18 | 33.74 |
| MolGAN ($\lambda_M = 0.5$) | 7.03 | 0.81 | 10.60 | 3.46 | 3.68 | 3.79 | **136.35** | 15.42 | 181.14 |
| MolGAN ($\lambda_M = 0$) | 8.37 | 0.99 | 11.98 | 3.77 | 3.74 | 4.36 | **159.93** | 15.45 | 208.58 |
| MolGAN ($\lambda_M = 0$) with ORGAN | 8.41 | 0.99 | 11.05 | 3.83 | 3.31 | 4.43 | **158.47** | 13.28 | 203.77 |
| GD-WGAN-GP ($\gamma_{max} = 0.25$) | 13.70 | 0.99 | 11.38 | 4.21 | 3.82 | 4.74 | **160.15** | 19.17 | 218.16 |
| GD-WGAN-GP ($\gamma_{max} = 0.5$) | 13.68 | 0.98 | 11.30 | 4.17 | 3.82 | 4.66 | **158.65** | 19.34 | 216.58 |
| GD-WGAN-GP ($\gamma_{max} = 0.75$) | 13.66 | 0.97 | 11.33 | 4.22 | 3.86 | 4.69 | **158.33** | 19.44 | 216.49 |
| GD-WGAN-GP ($\gamma_{max} = 1$) | 11.83 | 0.84 | 9.01 | 3.63 | 2.89 | 4.06 | **134.26** | 15.84 | 182.35 |
| GD-WGAN-GP with ORGAN | 11.70 | 0.81 | 7.87 | 3.58 | 2.36 | 3.90 | **131.64** | 12.04 | 173.89 |

the two predictors are consistent, and that GD-WGAN-GP with ORGAN achieves the lowest binding affinity among all methods. In contrast, the MolGAN variants ($\lambda_M = 0$) yield the highest median binding affinity scores, suggesting that these sequences may not bind to HLA-A*0201 despite their high immunogenicity predicted by the IEDB predictor Vita et al. (2019).

## B.4 Comparison of the GD-WGAN-GP variants with LSTM and Transformer architecture in the generator

Different generator architectures were explored by replacing the convolutional layers of the CNN generator with one-directional Long Short-Term Memory (LSTM) layers Hochreiter & Schmidhuber (1997) or Transformer layers Vaswani et al. (2017). Specifically, in the LSTM generator, the five residual blocks in Table 3 were replaced with four LSTM layers with a hidden size of 128, and the 1D convolution at layer 9 was replaced with a fully connected layer. In the Transformer generator, the layer 9 in Table 3 was also replaced with a fully connected layer and the five residual blocks in Table 3 were replaced with two Transformer layers with an embedding dimension of 128, eight heads, rotary positional encoding (base = 10000) Su et al. (2024), root-mean-square layer normalization Zhang & Sennrich (2019), and no causal masking in the attention layer. The feed-forward network in each Transformer layer consists of a single fully connected layer with a Gaussian Error Linear Unit (GELU) activation function Hendrycks & Gimpel (2023) and a hidden size four times the embedding dimension. The number of layers for the both generators are selected to be similar to the CNN generator with 660,629 model parameters. The four-layer LSTM generator has 696,213 parameters and the two-layer Transformer generator has 563,861 parameters.

The predicted immunogenicity score and the binding affinity of the GD-WGAN-GP ($\gamma_{max} = 1$) and the GD-WGAN-GP woth ORGAN with LSTM or Transformer generator are shown in the gray boxes in Figure 5 and Figure 6, respectively. The results in Figure 5 shows that GD-WGAN-GP ($\gamma_{max} = 1$) with LSTM and GD-WGAN-GP with ORGAN with transformer achieve a median of immunogenicity score higher than 0.9 from the Deepimmuno-CNN Li et al. (2021), similar to GD-WGAN-GP ($\gamma_{max} = 1$) with the CNN generator, while the result in Figure 6 shows that GD-WGAN-GP ($\gamma_{max} = 1$) with LSTM achieve a much higher median binding affinity score with more than $10^4$ nM. The GD-WGAN-GP with ORGAN with transformer achieve a median binding affinity closer to that for the GD-WGAN-GP ($\gamma_{max} = 1$).

## B.5 Comparison of the ablated GD-WGAN-GP variants

In the ablation study, five GD-WGAN-GP variants are considered: GD-WGAN-GP ($\gamma_{max} = 1$), GD-WGAN-GP ($\gamma_{max} = 1$) without the reward network, GD-WGAN-GP ($\gamma_{max} = 1, S_{scale} = 1$), GD-WGAN-GP without the switch and ORGAN, and GD-WGAN-GP with ORGAN. Specifically, the GD-WGAN-GP ($\gamma_{max} = 1$) without the reward network is implemented by directly using the immunogenicity predictor for the generator, replacing $S$ with $P$.

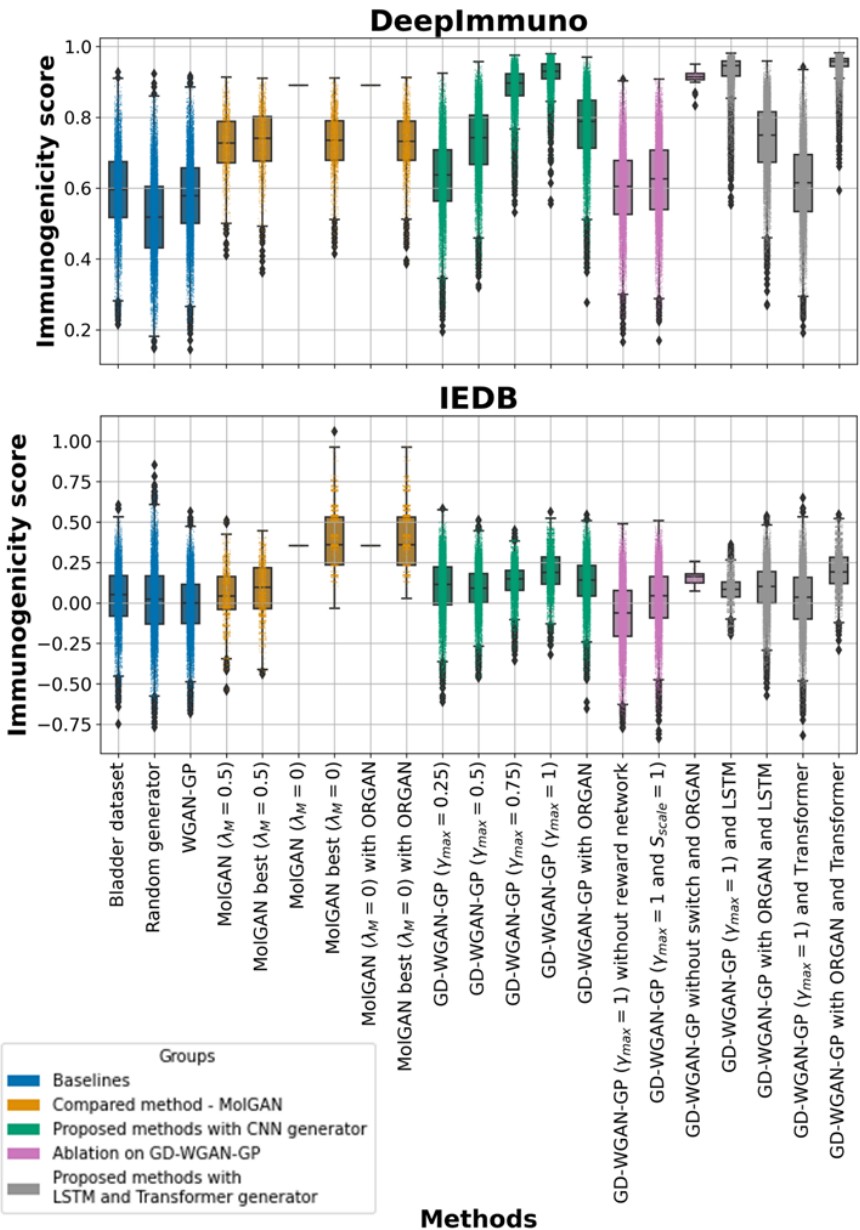

Figure 5: The dots and box plots showing the immunogenicity scores predicted by DeepImmuno-CNN Li et al. (2021) (top) and IEDB Vita et al. (2019) (bottom). The evaluated sequences are 9–10 mers without repetition from each method, the same as those evaluated for the second column in Table 1. The blue boxes represent the scores from the baselines and dataset, the orange boxes represent the scores from the MolGAN variants, the green boxes represent the scores from the GD-WGAN-GP variants, the pink boxes represent the scores from the ablated GD-WGAN-GP variants, and the gray boxes represent the scores from the GD-WGAN-GP variants with LSTM or Transformer architectures in the generator.

The average immunogenicity scores during training for these ablated variants are shown in Figure 7 (a). GD-WGAN-GP ($\gamma_{max} = 1$) and GD-WGAN-GP without the switch and ORGAN achieve average immunogenicity scores above 0.9 after 800 epochs. GD-WGAN-GP with ORGAN reaches an average score of around 0.8, while both GD-WGAN-GP ($\gamma_{max} = 1$) without the reward network and GD-WGAN-GP ($\gamma_{max} = 1$, $S_{scale} = 1$) fall below 0.7 after 600 epochs.

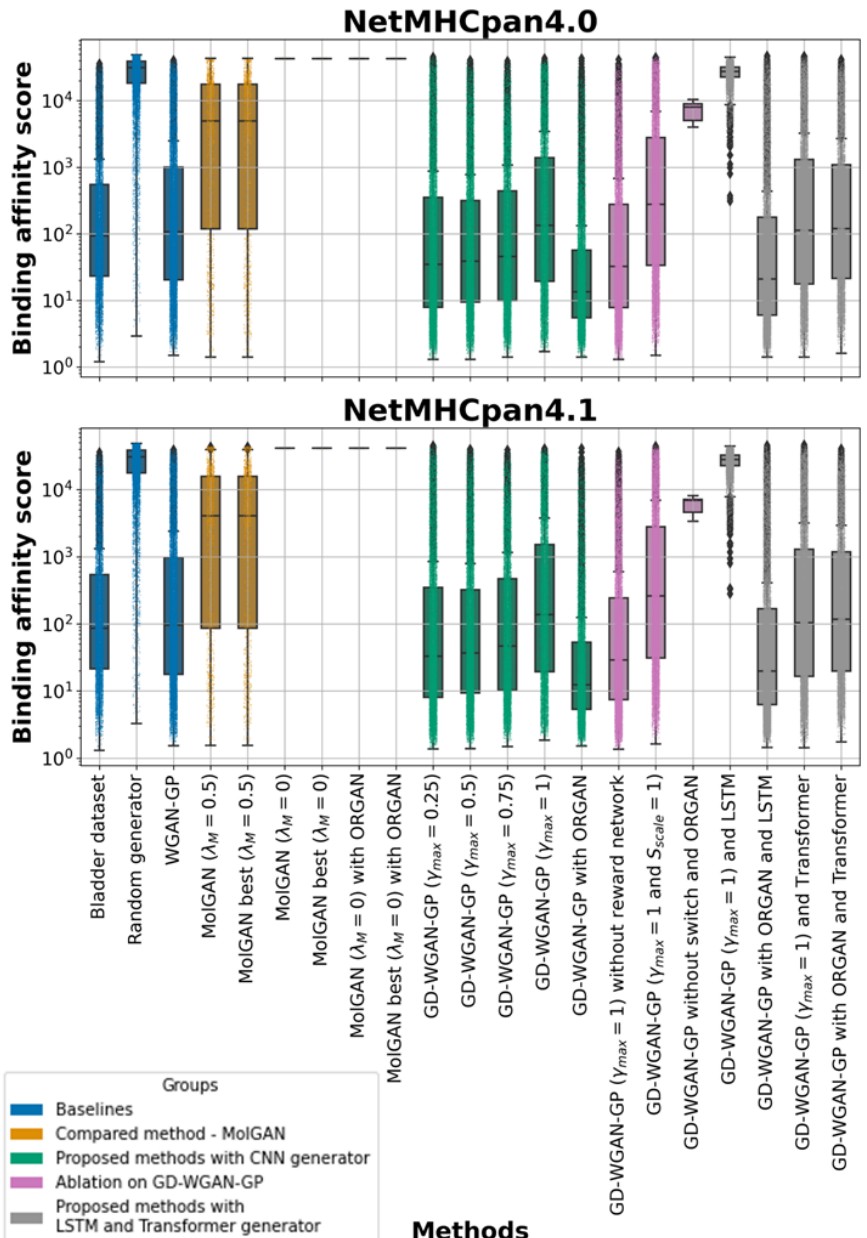

Figure 6: The dots and box plots showing the binding affinity (nM) predicted by NetMHCpan-4.0 Jurtz et al. (2017) (top) and NetMHCpan-4.1 Reynisson et al. (2020) (bottom). The evaluated sequences are 9–10 mers without repetition from each method. The blue boxes represent the scores from the baselines and dataset, the orange boxes represent the scores from the MolGAN variants, the green boxes represent the scores from the GD-WGAN-GP variants, the pink boxes represent the scores from the ablated GD-WGAN-GP variants, and the gray boxes represent the scores from the GD-WGAN-GP variants with LSTM or Transformer architectures in the generator.

The unique rate curves in Figure 7 (b) show that, despite the high average immunogenicity score ($>0.9$) of GD-WGAN-GP without the switch and ORGAN, its unique rate drops to nearly 0 after 100 epochs. In contrast, GD-WGAN-GP ($\gamma_{max} = 1$) maintains a unique rate of around 1.0 throughout training.

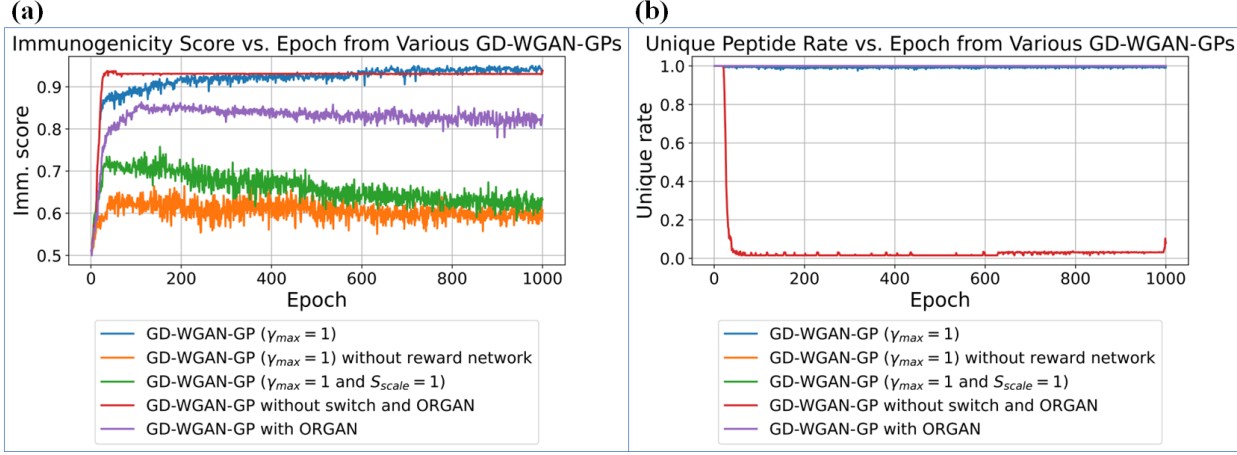

Figure 7: The line plot showing the predicted immunogenicity score from DeepImmuno-CNN Li et al. (2021) during training for different ablated variants of GD-WGAN-GP.

### B.6 Experiment on edit distance

The edit distance distributions for all considered methods are shown in Figure 8. The distributions are evaluated by computing the edit distance between each pair of generated sequences. The highest mean value is achieved by the random generator, with a mean of 9.00 in Figure 8 (a). As shown in Figure 8 (b), MolGAN (best, $\lambda_M = 0$) has a lower mean edit distance compared to MolGAN ($\lambda_M = 0.5$). Figure 8 (c) shows that GD-WGAN-GP with the switching mechanism tends to have lower diversity as $\gamma_{max}$ increases. Among the ablated variants of GD-WGAN-GP ($\gamma_{max} = 1$), the LSTM-based generator achieves the lowest mean edit distance compared to other ablated methods, as shown in Figure 8 (d).

### B.7 Analysis between the two considered datasets

The edit distance distribution between the bladder and brain datasets in Figure 9 shows that the peptide diversity in the two datasets is similar, with the bladder dataset having a median edit distance of 8.04 and the brain dataset a median edit distance of 7.96.

The dots and box plots comparing the bladder and brain datasets are shown in Figure 10. The median binding affinity of the brain dataset is lower than that of the bladder dataset, which may explain why the number of binders in Table 2 for bladder cancer is generally lower than in Table 9 for brain cancer.

## C Experiment using brain cancer epitopes

In this section, the comparison of the proposed GD-WGAN-GP with the generator trained using the architecture from Li et al. (2021) and MolGAN (Cao & Kipf, 2022) is presented in Table 8 using the 2,454 brain cancer epitopes from TSNAdb (Wu et al., 2018) with IC50 < 500nM. The network architecture and hyperparameters are the same as in Section 2.

The results in Table 8 demonstrate that the proposed GD-WGAN-GP with a switching mechanism enables users to balance the average immunogenicity score and the uniqueness of the generated peptides. These peptides also exhibit high binding affinity to HLA-A*0201, as shown in Table 9, outperforming those generated by MolGAN (Cao & Kipf, 2022). Among all models, the generator trained with the GD-WGAN-GP incorporating ORGAN's repetition penalty (Guimaraes et al., 2018) achieves the highest sum of average immunogenicity score and ratio of non-repeated peptides (1.7917), indicating it as the most effective design.

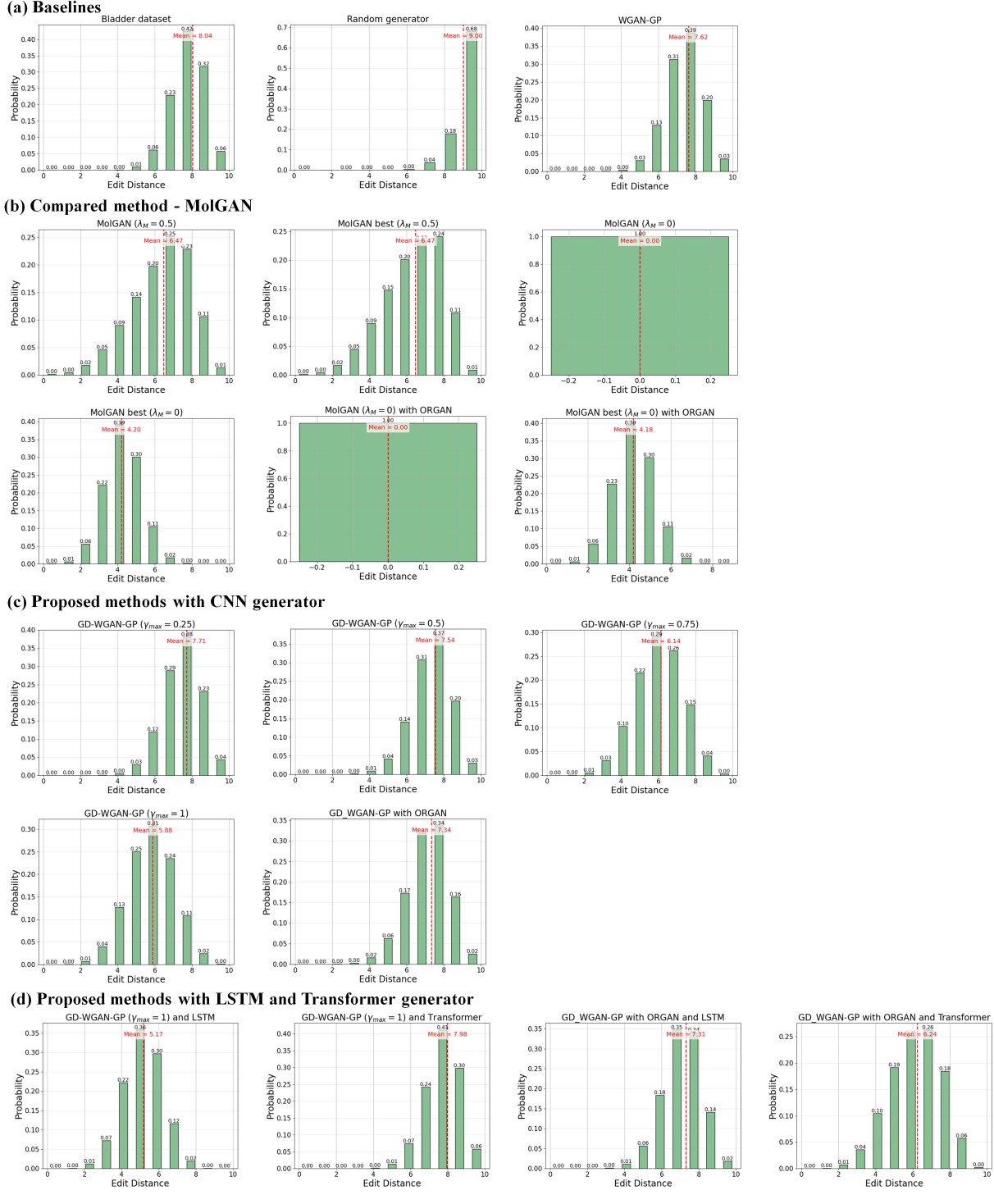

Figure 8: The edit distance distribution between different methods.

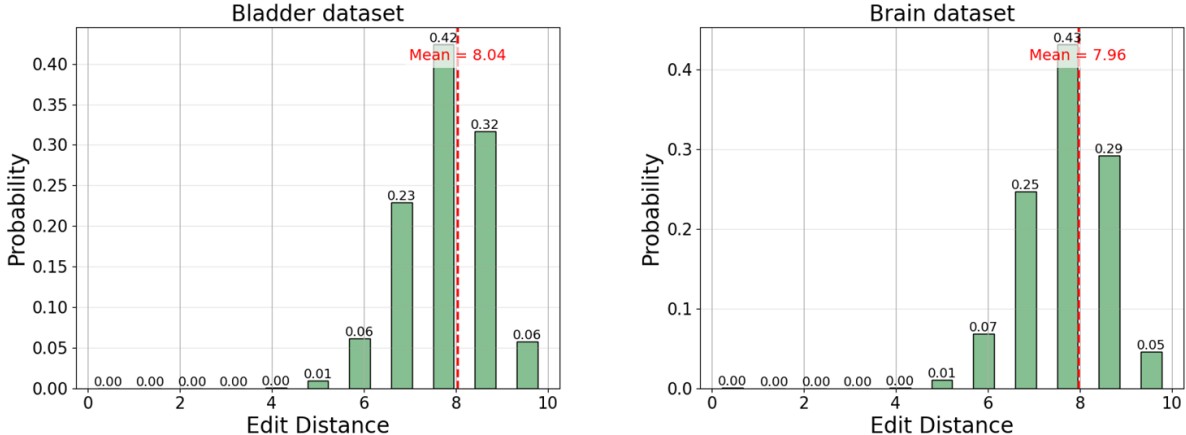

Figure 9: The edit distance distribution between the bladder dataset and the brain dataset used in this work.

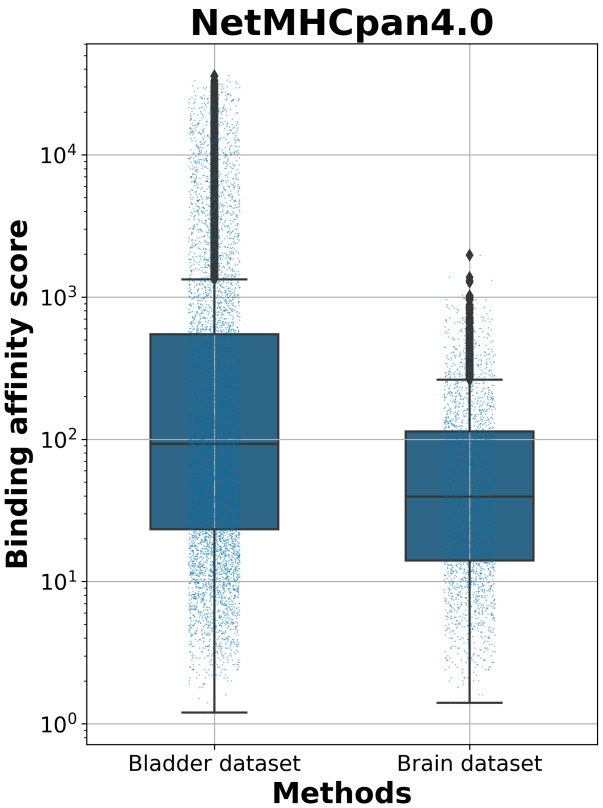

Figure 10: The dots and box plots between the bladder dataset and the brain dataset used in this work.

Table 8: Comparison of different designs of GANs trained to generate peptide vaccine candidates for brain cancer for 1000 epochs except for MolGAN$^{\text{best}}$ (Cao & Kipf, 2022), where the generator checkpoint is selected from the epoch that yields the highest sum of the immunogenicity score (imm. score) and the ratio of non-repeated peptides among a batch of 64 generated samples across all 1000 epochs. The imm. score is predicted by the predictor from Li et al. (2021) and its average is across the generated non-repeated peptides after removing peptides more than 10-mer and less than 9-mer. The percentage (percent.) of non-repeated peptides is the percentage of the value of the number of non-repeated peptides divided by the number of generated peptides (10,000). "min." indicates minute. In the Algorithm column, "with ORGAN" indicates the reward for each generated data is penalized by divided the number of its repetition in a batch during training (Guimaraes et al., 2018).

| Algorithm | Average imm. score | Percent. of peptides with 9-10-mer (%) | Percent. of non-repeated peptides (%) | Total training time (min.) |
|---|---|---|---|---|
| WGAN-GP | 0.57 | 99.85 | **99.41** | 12.80 |
| MolGAN ($\lambda_M = 0.5$) | 0.76 | 95.56 | 10.28 | 70.90 |
| MolGAN$^{\text{best}}$ ($\lambda_M = 0.5$) | 0.68 | 99.98 | 73.61 | 76.14 |
| MolGAN ($\lambda_M = 0$) | 0.89 | 100.00 | 0.01 | 70.82 |
| MolGAN$^{\text{best}}$ ($\lambda_M = 0$) | 0.73 | 100.00 | 22.30 | 73.91 |
| MolGAN ($\lambda_M = 0$) with ORGAN | 0.89 | 100.00 | 0.01 | 70.71 |
| MolGAN$^{\text{best}}$ ($\lambda_M = 0$) with ORGAN | 0.73 | 100.00 | 22.85 | 71.15 |
| GD-WGAN-GP ($\gamma_{max} = 0.25$) | 0.64 | 97.99 | 98.88 | 64.83 |
| GD-WGAN-GP ($\gamma_{max} = 0.5$) | 0.82 | 99.86 | 97.04 | 87.53 |
| GD-WGAN-GP ($\gamma_{max} = 0.75$) | 0.86 | 99.94 | 76.77 | 75.70 |
| GD-WGAN-GP ($\gamma_{max} = 1$) | **0.90** | 99.86 | 60.15 | 76.88 |
| GD-WGAN-GP with ORGAN | 0.83 | 99.96 | 96.17 | 74.66 |

Table 9: Comparison of the predicted binding affinity between different designs of GANs trained to generate peptide vaccine candidates for brain cancer after training for 1000 epochs except for MolGAN[best] (Cao & Kipf, 2022), where the generator checkpoint is selected from the epoch that yields the highest sum of the immunogenicity score and the ratio of non-repeated peptides among a batch of 64 generated samples across all 1000 epochs. The percentage (percent.) of strong binders is the percentage of the ratio between the number of generated unique sequences having $IC_{50} < 500nM$ binding with HLA-A*0201 predicted by NetMHCpan v4.0 (Jurtz et al., 2017) and the number of generated peptides (10,000). In the Algorithm column, "with ORGAN" indicates the reward for each generated data is penalized by divided the number of its repetition in a batch during training (Guimaraes et al., 2018).

| Algorithm | Percent. of unique binders (%) | Number of generated unique peptides with 9-10-mer | Number of generated peptides in the training set |
|---|---|---|---|
| WGAN-GP | **93.79** | **9796** | 0 |
| MolGAN ($\lambda_M = 0.5$) | 6.34 | 998 | 0 |
| MolGAN[best] ($\lambda_M = 0.5$) | 14.72 | 7352 | 0 |
| MolGAN ($\lambda_M = 0$) | 0.00 | 1 | 0 |
| MolGAN[best] ($\lambda_M = 0$) | 0.00 | 2225 | 0 |
| MolGAN ($\lambda_M = 0$) with ORGAN | 0.00 | 1 | 0 |
| MolGAN[best] ($\lambda_M = 0$) with ORGAN | 0.00 | 2281 | 0 |
| GD-WGAN-GP ($\gamma_{max} = 0.25$) | 89.45 | 9686 | 0 |
| GD-WGAN-GP ($\gamma_{max} = 0.5$) | 88.37 | 9690 | 0 |
| GD-WGAN-GP ($\gamma_{max} = 0.75$) | 72.10 | 7671 | 0 |
| GD-WGAN-GP ($\gamma_{max} = 1$) | 51.89 | 6001 | 0 |
| GD-WGAN-GP with ORGAN | 89.29 | 9613 | 0 |

