# OpenReview forum: "Epitope Generation for Peptide-based Cancer Vaccine using Goal-directed Wasserstein Generative Adversarial Network with Gradient Penalty"
_TMLR — Rejected by TMLR_

### Review · Reviewer_KBo8 · 2025-08-02

**Summary Of Contributions:**

- This paper improves WGAN-GP for peptide generation. The main difference is the addition of goal-directed terms from immunogenicity/binding predictors.
- Introduce a switch mechanism to discourage rewarding duplicated sequences.

**Audience:**

Yes

**Audience Explanation:**

Yes, biological sequence generation is an interesting application problem area.

**Claims And Evidence:**

No

**Claims Explanation:**

**Strengths:**
- The framework is a logical extension of WGAN-GP.
- Diagram is illustrative of the entire proposed framework GD-WGAN-GP and the training recipe is clear.
- On peptide generation setting, introducing gradient guidance improved the property of sequence generation. It also outperformed MolGAN.
- The proposed switching mechanism reduced number of duplicated sequences.

**Weakness:**
- Diversity beyond de duplication is not evaluated. No edit-distance/entropy/position-wise diversity.
- See more questions below - that will help understanding of the improved results better.

**Requested Changes:**

**Questions:**
- Can the author clarify the difference with WGAN-GP from Li et al. 2021? Is the main difference GD-WGAN-GP introduced a reward module?

- How does GD-WGAN-GP differ from MolGAN? Is the main difference in the use of gradient penalty?

- Confused about why the reward network needs to be trained during training of GD-WGAN_GP to predict immunogenicity score, when we already have an immunogenicity score? Is it because you need gradients from the reward module?

- Switching vs. ORGAN: if ORGAN already penalizes repeats, why also a hard switch? Show a clean ablation: reward on (no switch, no ORGAN) vs switch vs ORGAN

- Why is the training time for WGAN-GP much shorter than GD-WGAN-GP?

**Requested revisions:**
- Ablation on strategies on diversifying candidates.
- Add edit-distance/entropy metrics and show distributions (train vs. gen).

---

> ### Author Response · Authors · 2025-08-19
>
> ## Response to the Questions
>
> > Can the author clarify the difference with WGAN-GP from Li et al. 2021? Is the main difference GD-WGAN-GP introduced a reward module?
>
> No. The main differences are the introduction of the scaling factor, the switching mechanism, and the reward term in the generator’s loss function.
>
> > How does GD-WGAN-GP differ from MolGAN? Is the main difference in the use of gradient penalty?
>
> We think that the main difference lies in the weighting strategy between the critic output and the reward output in the generator’s loss function.
>
> > Confused about why the reward network needs to be trained during training of GD-WGAN_GP to predict immunogenicity score, when we already have an immunogenicity score? Is it because you need gradients from the reward module?
>
> We found that the generator failed to produce peptides with improved immunogenicity scores when the predictor was used directly, which is shown in the ablation study in Section B.5.
>
> > Switching vs. ORGAN: if ORGAN already penalizes repeats, why also a hard switch? Show a clean ablation: reward on (no switch, no ORGAN) vs switch vs ORGAN
>
> - There may be a misunderstanding that GD-WGAN-GP with ORGAN is implemented with the switching mechanism; however, GD-WGAN-GP with ORGAN applies only the ORGAN method without the switch.
> - As for the ablation study, five GD-WGAN-GP variants are considered: (1) GD-WGAN-GP ($\gamma_{max}=1$), (2) GD-WGAN-GP ($\gamma_{max}=1$) without the reward network, (3) GD-WGAN-GP ($\gamma_{max}=1$, $S_{scale}=1$), (4) GD-WGAN-GP without the switch and ORGAN, and (5) GD-WGAN-GP with ORGAN. The discussion of these variants is provided in Section B.5 of the revised manuscript. The average immunogenicity scores and unique rates of these methods during training are shown in Figure 7. The dots and box plots of the predicted immunogenicity and binding affinity of the ablated variants are shown in the gray boxes in Figures 5 and 6, respectively.
>
> > Why is the training time for WGAN-GP much shorter than GD-WGAN-GP?
>
> We have added the total computational time for each process during 1000 epochs of training for each method in Table 7 of the revised manuscript. The results show that the main increase in computational time arises from the forward pass of the immunogenicity predictor.
>
> ## Response to the Requested revisions
>
> > Ablation on strategies on diversifying candidates.
>
> The results in Figure 8 (c) shows that GD-WGAN-GP with the switching mechanism tends to have lower diversity as $\gamma_{max}$ increases.
>
> > Add edit-distance/entropy metrics and show distributions (train vs. gen).
>
> We have added the edit distance distribution of the generated peptides from different models and the dataset in Figure 8 of the revised manuscript.

---

### Review · Reviewer_6D2Z · 2025-08-03

**Summary Of Contributions:**

This manuscript presents a novel goal-directed WGAN-GP (GD-WGAN-GP) approach for generating immunogenic peptide candidates, targeting HLA-A*0201 for cancer vaccine development. The authors introduce a dual-reward system: a critic network for realism and a reward network (based on immunogenicity prediction) to steer the generator toward biologically promising peptides. Two diversity-promoting strategies—a switching mechanism and an ORGAN-inspired penalty—are evaluated to reduce sequence repetition. Experimental results on bladder and brain cancer peptide datasets show that their model can outperform standard GAN baselines in immunogenicity and uniqueness, although some trade-offs emerge. All code and data are shared for reproducibility.

**Additional Comments:**

It is unclear how the peptide generation process handles cases where multiple amino acids have the highest predicted probabilities at a given position.

**Audience:**

Yes

**Audience Explanation:**

TMLR readers interested in machine learning for biology, especially protein/peptide sequence generation, will find this work relevant. The results, even with their caveats, will be instructive for others developing generative models in low-data or high-stakes biomedical domains.

**Broader Impact Concerns:**

1. Reward hacking/dual use: The paper should explicitly discuss the risk that models might exploit weaknesses of in silico predictors, producing peptides that look good computationally but are biologically irrelevant or, in rare cases, harmful.
2. Potential for misuse: While low risk, generative biosequence models could be misapplied. This should be acknowledged in the impact statement.

**Claims And Evidence:**

No

**Claims Explanation:**

No. While the authors state that “this paper presents various designs of goal-directed Wasserstein Generative Adversarial Networks with Gradient Penalty (GD-WGAN-GP) for training generators capable of producing 9- to 10-mer peptide sequences with high predicted immunogenicity, low repetition rate, and strong binding affinity,” I find the results less than fully convincing for the following reasons:

1. Limited Baselines: The authors benchmark their methods only against other GAN-based models. However, several widely used non-GAN approaches in this domain—such as VAEs, language model-based generators, and genetic algorithms—are missing. Even simple baselines like generating random peptides from amino acid distributions or sampling directly from the training set would provide useful context.

2. Hyperparameter Choices for Baselines: The authors set the hyperparameter c in MolGAN to 0.01, but this WGAN variant is known to be sensitive to this parameter. There is no evidence that this is the optimal setting for MolGAN, which may disadvantage this baseline.

3. Evaluation Dependent on In Silico Models: All evaluation relies entirely on computational models for immunogenicity and binding affinity, with no wet-lab validation.

4. Limited Training Data: The training dataset used is relatively small, which may constrain the model’s ability to generalize.

5. Risk of Reward Model Overfitting: The reward network is trained to mimic the same immunogenicity model that is used for evaluation. This creates a real risk that the generator could overfit or exploit patterns in the predictor, rather than learning more general biological features—a phenomenon known as “reward hacking” or “gaming the predictor.” This risk is not sufficiently discussed.

6. Limited Generalizability Across Datasets: In the brain cancer peptide generation task, the proposed method shows lower binding affinity and diversity compared to bladder cancer, especially at higher γ values (greater emphasis on immunogenicity). This suggests that the model’s advantages may not extend to other datasets and highlights trade-offs between immunogenicity, diversity, and binding.

7. Clarity on Evaluation Procedures: It is unclear whether immunogenicity and binding affinity are evaluated using the same set of generated sequences from the same batch, or from separate batches. Greater clarity on this point would aid in reproducibility and interpretation.

**Requested Changes:**

1. Clarify overfitting risk and evaluation limitations: The manuscript must explicitly discuss the potential for overfitting or reward hacking due to reliance on a single predictor for both training and evaluation.
2. Discuss limitations observed in brain cancer results: The lower binding affinity and diversity should be clearly described and analyzed. The manuscript should address why this might happen and what can be done about it (e.g., further parameter tuning or multi-objective optimization).
3. Add more baselines: Include at least one non-GAN generative baseline (e.g., VAE, Transformer/LM-based peptide generator, or a random generation baseline) to better contextualize the gains from the proposed approach.
4. Evaluate generated sequences with additional, independent immunogenicity and binding affinity predictors, if possible, to check for overfitting and model exploitation.
5. Diversity analysis: Show sequence logos, amino acid distributions, or clustering to assess the biological relevance of the diversity achieved.
6. Clarify whether evaluation is done on single or multiple runs, and whether the same or separate sets of generated sequences are used for immunogenicity and binding affinity evaluations.
7. Expand the Broader Impact statement: Discuss dual-use and safety concerns, and clarify the limitations of in silico validation.

---

> ### Author Response · Authors · 2025-08-19
>
> ## Response to the Requested Changes
>
> > 1. Clarify overfitting risk and evaluation limitations: The manuscript must explicitly discuss the potential for overfitting or reward hacking due to reliance on a single predictor for both training and evaluation.
>
> We acknowledge the potential risk of overfitting or reward hacking when relying on a single predictor. This issue could be mitigated by introducing a regularization term in the optimization function, such as an L1- or L2-norm. Another possible strategy is to construct a validation dataset and apply early stopping if the validation loss increases during training. Additionally, incorporating multiple predictors and optimizing based on the combination of their outputs could further reduce overfitting.
>
> We added the discussion as the first paragraph of the limitation section, placed before the conclusion.
>
> > 2.	Discuss limitations observed in brain cancer results: The lower binding affinity and diversity should be clearly described and analyzed. The manuscript should address why this might happen and what can be done about it (e.g., further parameter tuning or multi-objective optimization).
>
> We acknowledge the reviewer’s comment regarding the limitations observed in the brain cancer results. As shown in Figure 10 of Appendix B, the higher percentage of non-repeated peptides among the generated brain cancer epitopes compared to the bladder cancer epitopes may be attributed to the lower median binding affinity of the brain dataset.
>
> A possible way to address this limitation is to incorporate a binding affinity predictor into our framework, similar to the integration of the immunogenicity predictor. However, this approach depends on the availability and reliability of suitable predictors.
>
> We have discussed it at the end of Section 3.2.1 and added the discussion in the second paragraph of the limitation section.
>
> > 3.	Add more baselines: Include at least one non-GAN generative baseline (e.g., VAE, Transformer/LM-based peptide generator, or a random generation baseline) to better contextualize the gains from the proposed approach.
>
> We have added a random generation baseline, implemented by uniformly sampling the sequence length (9 or 10) and then uniformly sampling one of the characters (ARNDCQEGHILKMFPSTWYV) for each position. Its predicted immunogenicity, binding affinity, and edit distance are shown in Figures 5, 6, and 8(a), respectively, of the revised manuscript.
>
> > 4.	Evaluate generated sequences with additional, independent immunogenicity and binding affinity predictors, if possible, to check for overfitting and model exploitation.
>
> We have included the prediction results from IEDB for immunogenicity and from NetMHCpan-4.1 for binding affinity at the bottom of Figures 5 and 6, respectively, of the revised manuscript.
>
> The results from the IEDB predictor show that the GD-WGAN-GP variants with CNN generators achieve higher immunogenicity scores compared to the random generator baseline, the training dataset, and WGAN-GP. The results from NetMHCpan-4.1 are consistent with those from NetMHCpan-4.0, indicating that GD-WGAN-GP with ORGAN achieves the lowest median binding affinity.
>
> > 5.	Diversity analysis: Show sequence logos, amino acid distributions, or clustering to assess the biological relevance of the diversity achieved.
>
> We have added the edit distance distribution of the generated peptides from all considered models and the dataset in Figure 8 of the revised manuscript and discussed it in Appendix B.6.
>
> > 6.	Clarify whether evaluation is done on single or multiple runs, and whether the same or separate sets of generated sequences are used for immunogenicity and binding affinity evaluations.
>
> - The evaluation is done in a single run, where each generator produces 10,000 peptide sequences using the same noise matrix. Each row of the matrix represents a batch (10,000 rows in total), and each column corresponds to a dimension of the noise vector (128 columns in total).
> - Yes, the same set of generated sequences is used for immunogenicity and binding affinity evaluations.
>
> We have detailed the generation process in the paragraph before Section 3.1.
>
> > 7.	Expand the Broader Impact statement: Discuss dual-use and safety concerns, and clarify the limitations of in silico validation.
>
> We have rewritten the latter part of the impact statement to include the limitation of in silico validation.
>
> ## Response to the Additional Comments
>
> The amino acid at each position is selected using the argmax function from the NumPy library. If multiple amino acids have the same highest probability, the amino acid symbol that appears leftmost in the sequence, “'ARNDCQEGHILKMFPSTWYV-'”, will be chosen.

---

> > ### Comment · Reviewer_6D2Z · 2025-08-29
> >
> > Thank you for carefully addressing my concerns in the revision. The manuscript is now much clearer and improved. I just have one small comment: on page 11, the sentence “The GD-WGAN-GP with ORGAN’s repetition penalty shows that the generated peptides have a similar property to the training set (binders to HLA-A0201) compared to WGAN-GP and .”* seems incomplete. You may want to revise this for clarity.

---

### Review · Reviewer_A9CS · 2025-08-04

**Summary Of Contributions:**

### Summary
The authors present a model for goal-directed peptide generation based on a Wasserstein GAN with gradient penalty. The gradient penalty is introduced to mitigate mode collapse during training and to encourage the generation of diverse peptide sequences. The model is trained on epitope sequences from the TSNAdb dataset, specifically for bladder and brain cancer, with the aim of learning a distribution over peptides with high binding affinity, while being also conditioned to generate sequences with enhanced immunogenicity. At inference time, the model is expected to generate diverse peptides satisfying both properties.

### Strengths
(S1 – Strong results aligned with the goal of the study): The model clearly outperforms the baselines by generating peptide sequences that exhibit both high binding affinity (Table 2) and high immunogenicity (Table 1). In contrast, the baseline models achieve strong results in only one of the two properties.

(S2 – Clear articulation of the problem): The manuscript effectively motivates the task and explains the relevance of optimizing peptide sequences for both properties. The objective — combining distributional learning with goal-directed generation to preserve one property while optimizing the other — is well communicated.

### Weaknesses
(W1 – Presentation of results lacks robustness):
* The results are not statistically supported. The absence of error bars and statistical significance tests raises concerns about the robustness of the reported differences. The authors should include standard deviations across multiple runs and apply statistical testing to validate their claims.
* While the number of unique generated sequences is reported — as a proxy for measuring mode collapse — this alone may not be sufficient. A more practical metric would be the computational cost (e.g., wall-clock time or flops) required to obtain a given number of unique valid samples. Overlap in generated sequences is not necessarily problematic if the method is efficient enough to explore the space through repeated sampling.

(W2 – Lack of clarity regarding the immunogenicity predictor vs. reward model):
The reward model is trained to mimic the immunogenicity predictor on both the training and generated peptides (see Equation 7). However, it is unclear why a surrogate model is used instead of directly applying the predictor. Presumably, the predictor simulates a real-world assay and is not available at inference, but this assumption should be clarified. A clearer explanation of the role of each component (predictor vs. reward model) would improve the reader’s understanding.

(W3 – Limited baseline comparisons):
Only two baselines are considered: MolGAN and WGAN-GP. MolGAN does not do gradient penaltization and WGAN-GP does not do goal-directed optimization. While this provides a useful ablation analysis, a more comprehensive comparison with state-of-the-art methods would strengthen the claims. Many goal-directed sequence generation approaches have been proposed in the context of ML-guided directed evolution (e.g., [1, 2, 3]), and these should be discussed or compared.

[1] Angermueller, Christof, et al. Population-based black-box optimization for biological sequence design. ICML, 2020.

[2] Kirjner, Andrew, et al. Improving protein optimization with smoothed fitness landscapes. arXiv:2307.00494, 2023.

[3] Bryant, Patrick, and Arne Elofsson. Peptide binder design with inverse folding and protein structure prediction. Communications Chemistry 6.1 (2023): 229.

### Questions
* Why is it necessary to optimize for high binding affinity if immunogenicity already implies that the peptide elicits an immune response? Wouldn't high immunogenicity typically depend on binding?
* The use of a CNN-based architecture is reasonable. However, how would other architectures — such as Transformer-based or RNN-based models [3] — perform in this setting?
* Goal-directed generators are often known to exploit weaknesses of the scoring function (predictor) rather than improving actual properties (see [4]). Is it possible that the generator here has simply learned to exploit blind spots of the immunogenicity predictor rather than generating truly improved peptides?

[4] Renz, Philipp, et al. On failure modes in molecule generation and optimization. Drug Discovery Today: Technologies 32 (2019): 55–63.

**Additional Comments:**

Minor: Figure 2 illustrates one-hot encoding, which is standard textbook knowledge. Since the concept is well-known to the target audience, the figure seems unnecessary and takes up considerable space. Consider removing it or relocating it to the appendix.

**Audience:**

Yes

**Audience Explanation:**

Goal-directed peptide generation is an important research direction. The authors demonstrate that their model can preserve peptide binding affinity and maintain sequence diversity while optimizing for immunogenicity.

However, the overall relevance of the findings is reduced due to several factors. First, it remains unclear whether alternative approaches, particularly those not based on GANs, might outperform the proposed model (W3). Second, the lack of error bars or statistical significance testing raises concerns about the robustness of the reported results (W1). Finally, it is not discussed whether the generator might have simply learned to exploit weaknesses in the immunogenicity predictor rather than producing genuinely improved sequences (see Questions).

**Broader Impact Concerns:**

no concerns.

**Claims And Evidence:**

No

**Claims Explanation:**

The authors aim to design a model that balances sequence diversity, preservation of binding affinity, and optimization of immunogenicity. The presented results suggest that the model is capable of achieving this balance. However, due to the limitations outlined in (W1)–(W3), the significance and robustness of these results remain uncertain are not significant yet.

**Requested Changes:**

* Include error bars and conduct statistical significance tests to support the robustness of the results (see W1).
* Extend the baseline comparison to include a broader range of state-of-the-art methods beyond WGAN-GP and MolGAN (see W3).
* If the concern about the model potentially exploiting weaknesses in the predictor is valid (see Questions), consider evaluating generated sequences using an external immunogenicity predictor that the model has not seen during training.

---

> ### Author Response · Authors · 2025-08-19
>
> ## Response to the Questions
>
> > Why is it necessary to optimize for high binding affinity if immunogenicity already implies that the peptide elicits an immune response? Wouldn't high immunogenicity typically depend on binding?
>
> The high-binding-affinity peptides in the training dataset help guide the generator to produce sequences with similar properties. One reason we evaluate the generated peptides by binding affinity is to verify that they preserve this property of the training dataset while optimizing immunogenicity. In addition, although immunogenicity depends partly on binding, relying on a single immunogenicity predictor may introduce bias. For example, as shown in the comparison between GD-WGAN-GP ($\gamma_{max}=1$) and GD-WGAN-GP with ORGAN, peptides generated by GD-WGAN-GP ($\gamma_{max}=1$) achieve higher immunogenicity scores from DeepImmuno-CNN but also exhibit higher binding affinity than those from GD-WGAN-GP with ORGAN, demonstrating that predicted immunogenicity alone does not guarantee lower binding affinity.
>
> > The use of a CNN-based architecture is reasonable. However, how would other architectures — such as Transformer-based or RNN-based models [3] — perform in this setting?
>
> We have tested generators with architectures based on one-directional LSTM and Transformer, as detailed in Section B.4 of the revised manuscript. We found that the Transformer generator trained with GD-WGAN-GP and ORGAN achieves a higher immunogenicity score compared to the CNN generator, but it also has a higher binding affinity. The LSTM generator trained with GD-WGAN-GP and the switching mechanism achieves a median immunogenicity score above 0.9, but its binding affinity exceeds $10^4$.
>
> > Goal-directed generators are often known to exploit weaknesses of the scoring function (predictor) rather than improving actual properties (see [4]). Is it possible that the generator here has simply learned to exploit blind spots of the immunogenicity predictor rather than generating truly improved peptides?
>
> To check if the generator is producing truly improved peptides, we have included the prediction results from IEDB for immunogenicity and from NetMHCpan-4.1 for binding affinity at the bottom of Figures 5 and 6, respectively, of the revised manuscript.  The results from the IEDB predictor show that the GD-WGAN-GP variants with CNN generators achieve higher immunogenicity scores compared to the random generator baseline, the training dataset, and WGAN-GP. The results from NetMHCpan-4.1 are consistent with those from NetMHCpan-4.0, indicating that the GD-WGAN-GP CNN variants achieve lower binding affinity compared to the random generator and MolGAN, with GD-WGAN-GP with ORGAN achieving the lowest median binding affinity.
>
> ## Response to the Requested Changes
>
> > Include error bars and conduct statistical significance tests to support the robustness of the results (see W1).
>
> We have included the dots and box plots for the prediction results of immunogenicity and binding affinity in Figures 5 and 6, respectively, of the revised manuscript. We also added the standard deviation in Table 1.
>
> > Extend the baseline comparison to include a broader range of state-of-the-art methods beyond WGAN-GP and MolGAN (see W3).
>
> In the framework of our architecture, we have extended to the most recent and sophisticated generative AI techniques such as the attention-based Transformer and the RNN-based LSTM. We focus on model-based generation approaches; model-free approaches are beyond the scope of this work.
>
> > If the concern about the model potentially exploiting weaknesses in the predictor is valid (see Questions), consider evaluating generated sequences using an external immunogenicity predictor that the model has not seen during training.
>
> We have included the prediction results from IEDB for immunogenicity at the bottom of Figures 5 of the revised manuscript.
>
> ## Response to the Additional Comments
>
> > Minor: Figure 2 illustrates one-hot encoding, which is standard textbook knowledge. Since the concept is well-known to the target audience, the figure seems unnecessary and takes up considerable space. Consider removing it or relocating it to the appendix.
>
> We have moved Figure 2 in the original version to Appendix A in the revised manuscript.

---

### Decision · Action_Editor_Jg2z · 2025-09-18

**Recommendation:** Reject

**Additional Comments:**

I suggest major revision (rejection with possibility to resubmit). Possibility of reward hacking need to be addressed, by for example using multiple independently trained immunogenicity predictors, different independent scoring methods. Also increasing the number of baselines can help demonstrate that the method is competitive with SOTA.

**Audience:**

Yes

**Audience Explanation:**

There is agreement between the reviewers  that TMLR's audience would be interested in knowing the findings of this paper

**Claims And Evidence:**

No

**Claims Explanation:**

Unfortunately Reviewer A9CS is right to point out that reward hacking is quite likely in part responsible for the results. Looking at Figure 5 in the final manuscript and using the IEDB score as the one most likely unbiassed, the compared method outperforms the presented one.

**Resubmission Of Major Revision:**

The authors may consider submitting a major revision at a later time.